# Molecular mechanism of Mad1 kinetochore targeting by phosphorylated Bub1

Elyse S Fischer[1] (ID), Conny W H Yu[1] (ID), Dom Bellini[1] (ID), Stephen H McLaughlin[1] (ID), Christian M Orr[2] (ID), Armin Wagner[2], Stefan M V Freund[1] & David Barford[1,*] (ID)

## Abstract

During metaphase, in response to improper kinetochore-microtubule attachments, the spindle assembly checkpoint (SAC) activates the mitotic checkpoint complex (MCC), an inhibitor of the anaphase-promoting complex/cyclosome (APC/C). This process is orchestrated by the kinase Mps1, which initiates the assembly of the MCC onto kinetochores through a sequential phosphorylation-dependent signalling cascade. The Mad1-Mad2 complex, which is required to catalyse MCC formation, is targeted to kinetochores through a direct interaction with the phosphorylated conserved domain 1 (CD1) of Bub1. Here, we present the crystal structure of the C-terminal domain of Mad1 (Mad1$^{CTD}$) bound to two phosphorylated Bub1$^{CD1}$ peptides at 1.75 Å resolution. This interaction is mediated by phosphorylated Bub1 Thr461, which not only directly interacts with Arg617 of the Mad1 RLK (Arg-Leu-Lys) motif, but also directly acts as an N-terminal cap to the CD1 α-helix dipole. Surprisingly, only one Bub1$^{CD1}$ peptide binds to the Mad1 homodimer in solution. We suggest that this stoichiometry is due to inherent asymmetry in the coiled-coil of Mad1$^{CTD}$ and has implications for how the Mad1-Bub1 complex at kinetochores promotes efficient MCC assembly.

**Keywords** Bub1; Cell cycle; Mad1; Mitotic checkpoint complex; Spindle assembly checkpoint

**Subject Categories** Cell Cycle; Structural Biology

## Introduction

Cell division, a process by which cells duplicate themselves, along with their DNA, is the most fundamental process of life. Faithful chromosome segregation requires surveillance by the SAC, which, in response to improper kinetochore-microtubule attachments, delays anaphase onset until biorientation of attachments has been achieved (Foley & Kapoor, 2013; Musacchio, 2015; Sacristan & Kops, 2015). SAC activation then triggers production of the mitotic checkpoint complex (MCC) consisting of BubR1, Bub3, Mad2 and Cdc20 (Sudakin *et al*, 2001; Chao *et al*, 2012). The MCC functions by binding and inhibiting the E3 ubiquitin ligase, the anaphase-promoting complex/cyclosome (APC/C), when the APC/C is bound by a second coactivating molecule of Cdc20 (Herzog *et al*, 2009; Izawa & Pines, 2015; Alfieri *et al*, 2016; Yamaguchi *et al*, 2016). Inhibition of the APC/C then delays premature chromosome segregation by preventing APC/C-mediated degradation of two key cell cycle regulators, cyclin B and securin (Cohen-Fix *et al*, 1996; Clute & Pines, 1999).

Understanding the molecular mechanisms of how kinetochores recruit checkpoint proteins and how each protein contributes to the localization and stimulation of downstream components remains a major unresolved question in the field. Recent work points towards hierarchical recruitment of SAC proteins to the outer kinetochore, by means of an Mps1-dependent phosphorylation cascade, which creates a catalytic platform for MCC assembly (Fig 1A and B) (Faesen *et al*, 2017; Ji *et al*, 2017b; Dou *et al*, 2019). This begins with Mps1 phosphorylating several MELT (methionine–glutamate–leucine–threonine) motifs on the outer kinetochore protein Knl1, which then recruits the Bub3-Bub1 complex (London *et al*, 2012; Shepperd *et al*, 2012; Yamagishi *et al*, 2012; Primorac *et al*, 2013; Vleugel *et al*, 2015). Next, Mps1 phosphorylates Bub1 at a central conserved domain 1 (CD1), which recruits the Mad1-Mad2 complex (London & Biggins, 2014a, 2014b; Ji *et al*, 2017b; Zhang *et al*, 2017). Although the precise recruitment pathway of Cdc20 for MCC formation is still debated, it is likely that Bub1, through ABBA and KEN motifs just C-terminal to the CD1 domain, plays a role in recruiting and repositioning Cdc20 close to the Mad1-Mad2 complex (Diaz-Martinez *et al*, 2015; Lischetti *et al*, 2015; Di Fiore *et al*, 2016; Zhang *et al*, 2019).

Recently, a third phosphorylation event catalysed by Mps1 was identified at the very C-terminus of Mad1, which significantly enhances rates of MCC formation, possibly by promoting an interaction with Cdc20 and repositioning the Mad2-interacting motif (MIM) of Cdc20 close to Mad2 (Ji *et al*, 2017a, 2017b). In the last step of MCC assembly, the Mad1-Mad2 complex acts as a platform for conversion of Mad2 in the open conformation (O-Mad2) into closed-Mad2 (C-Mad2) through a template conversion mechanism (Sironi *et al*, 2002; Luo *et al*, 2004; De Antoni *et al*, 2005; Mapelli *et al*,

---

1   MRC Laboratory of Molecular Biology, Cambridge, UK
2   Diamond Light Source Ltd, Didcot, UK
    *Corresponding author (lead contact). Tel: +44 01223 267075; Email: dbarford@mrc-lmb.cam.ac.uk

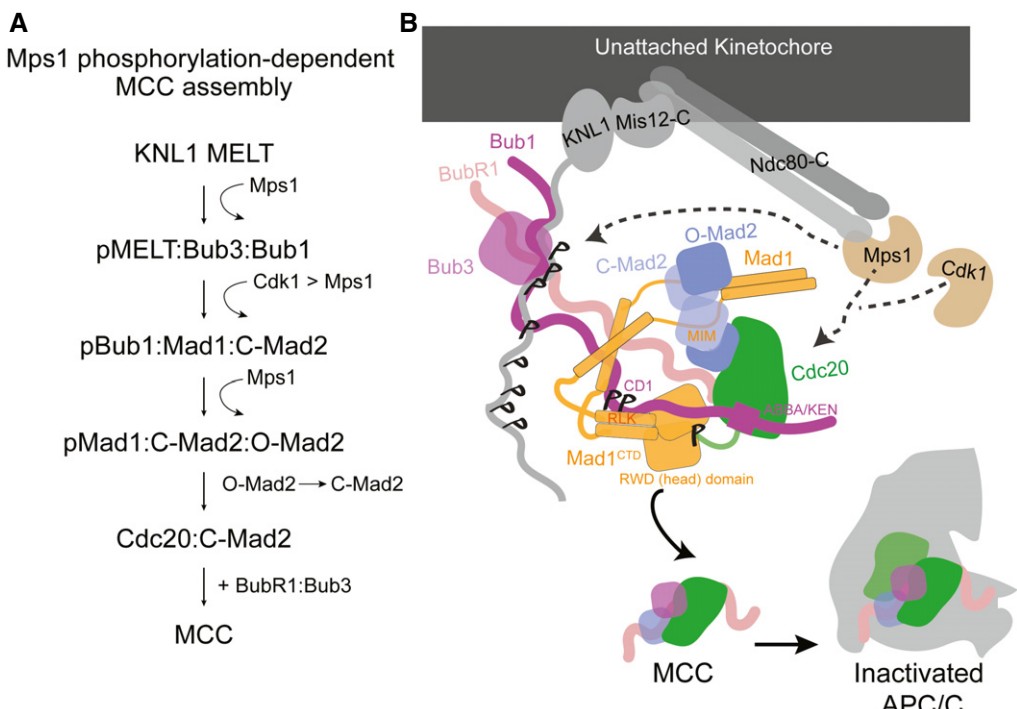

**Figure 1. Model of MCC assembly and the Bub1-Mad1-Mad2-Cdc20 complex at kinetochores.**

A  Outline of the essential sequential steps of MCC assembly onto the outer kinetochore.
B  A model of MCC assembly onto the outer kinetochore. MCC assembly occurs in a stepwise manner that is under the control of Mps1 and Cdk1 kinases. Several MELT motifs on Knl1 are first phosphorylated by Mps1. pMELTs then recruit the Bub3:Bub1 complex which is then phosphorylated first by Cdk1 then by Mps1 on the Bub1 CD1 domain. Phosphorylated Bub1 then recruits the Mad1:C-Mad2 complex which then acts as a platform for O-Mad2 binding and catalyst for conversion into C-Mad2. Bub1 is likely responsible for Cdc20 targeting to kinetochores through its KEN and ABBA motifs as well BubR1 through a central Bub1-BubR1 dimerization domain. The CTD of Mad1 is phosphorylated by Mps1 which interacts with an N-terminal tail of Cdc20. Bub1 therefore acts as a scaffold to position C-Mad2, BubR1 and Cdc20 in close proximity for efficient MCC formation. Once formed soluble MCC then binds and inactivates the APC/C, preventing metaphase to anaphase progression.

2007; Simonetta *et al*, 2009). C-Mad2 is then passed onto Cdc20 to form the Cdc20:C-Mad2 complex which binds BubR1 with high affinity to generate the MCC (Kulukian *et al*, 2009; Chao *et al*, 2012; Faesen *et al*, 2017). Altogether, this suggests that MCC assembly onto kinetochores requires a highly regulated Mps1-dependent phosphorylation cascade which utilizes Bub1 as a key platform for targeting and repositioning the catalytic engine Mad1:C-Mad2 as well as BubR1 and Cdc20.

The importance of the phosphorylated Bub1-Mad1 complex in SAC activation was suggested in 2000 in a study which also identified the RLK motif of Mad1 as being essential for Mad1 kinetochore association (Brady & Hardwick, 2000). The role of the Mad1 RLK motif and the Bub1 CD1 domain in SAC signalling and Mad1 kinetochore targeting has since been well studied, and both are highly conserved across species (Kim *et al*, 2012; Heinrich *et al*, 2014; London & Biggins, 2014a; Zhang *et al*, 2017; Luo *et al*, 2018). It was not until 2014 that a direct interaction between Bub1$^{CD1}$ and Mad1$^{CTD}$ in budding yeast was shown to be dependent on phosphorylation of Bub1$^{CD1}$ by Mps1 at Thr453 and Thr455 (London & Biggins, 2014a). In 2017, two independent studies confirmed the direct interaction of human Bub1$^{CD1}$-Mad1$^{RLK}$ which was dependent on phosphorylation at Ser459 and Thr461 (Ji *et al*, 2017b; Zhang *et al*, 2017). Schematics of human Bub1 and human Mad1 are shown in Fig 2A.

Because there was no structural information of the Bub1-Mad1 complex, the overall architecture and how phosphorylation

promotes specificity of the Bub1-Mad1 interaction remained unclear. Here, we present the first structure of the human Mad1$^{CTD}$ homodimer bound to two phosphorylated human Bub1$^{CD1}$ peptides at 1.75 Å. We found that this interaction is dependent on pThr461 which makes direct contact with Arg617 of the conserved Mad1 RLK motif, whereas pSer459 is not involved. We were able to gain further detailed structural insights into this interaction using NMR spectroscopy which suggests a strikingly global conformational change across Mad1$^{CTD}$ upon Bub1$^{CD1}$ binding. Additionally, and to our surprise, we discovered that in solution only one Bub1$^{CD1}$ peptide binds to the Mad1$^{CTD}$ homodimer. We hypothesize that this is due to an inherent asymmetry of Mad1$^{CTD}$, also apparent in the previously crystallized apo Mad1$^{CTD}$ homodimer (Kim *et al*, 2012). This asymmetry makes only one side of the Mad1$^{CTD}$ dimer favourable for binding and might have important implications for the function of the Mad1:Mad2:Bub1:Cdc20 complex at kinetochores.

## Results

### Overall architecture of Bub1$^{CD1}$ bound to Mad1$^{CTD}$

We determined a near-atomic resolution crystal structure of the human Mad1$^{CTD}$ homodimer (residues 597–718) bound to two doubly phosphorylated Bub1 peptides comprising the CD1 domain,

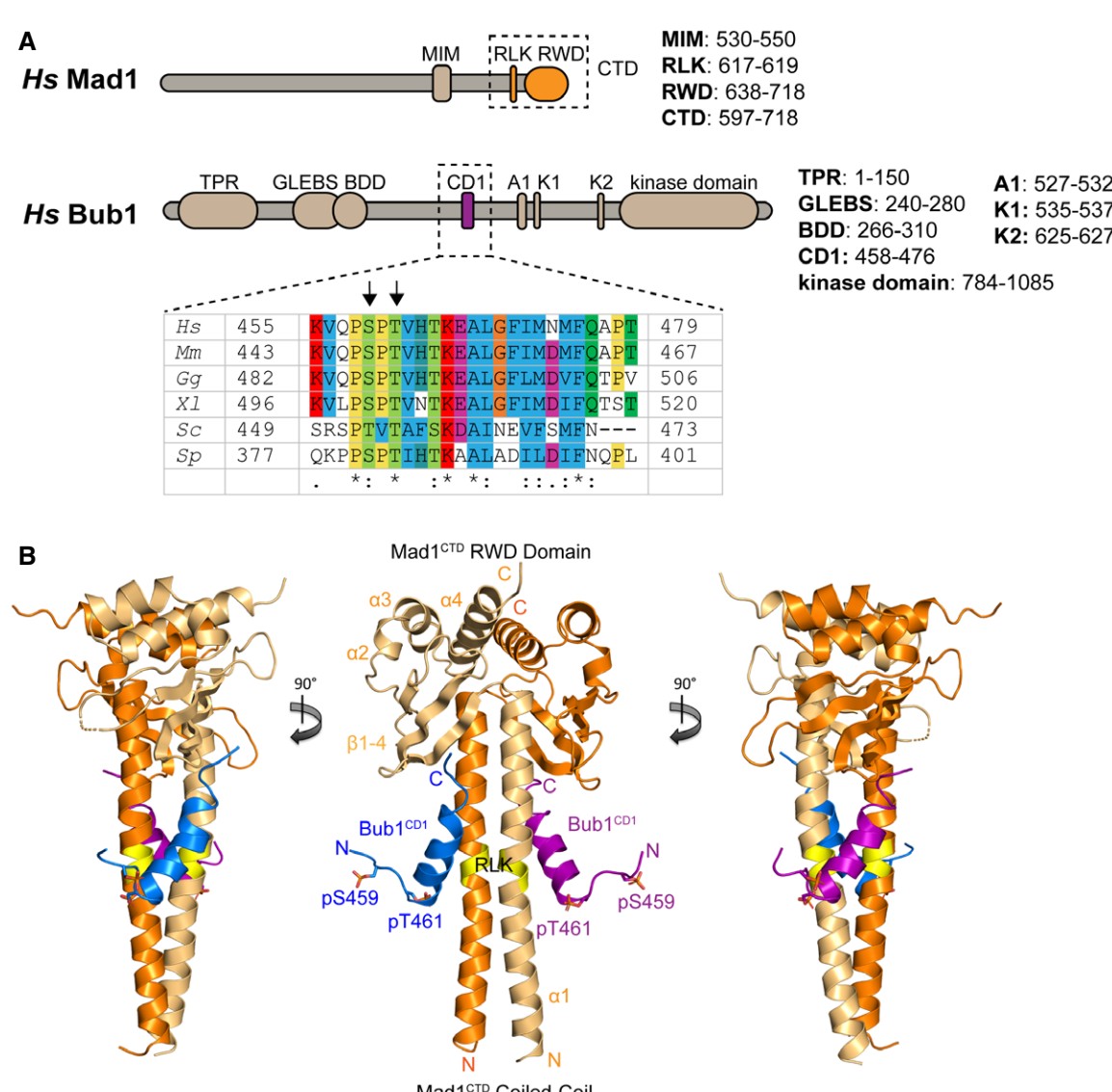

**Figure 2. Overview of the human Mad1^CTD-Bub1^CD1 complex.**

A  Schematics of full-length human Mad1 and Bub1. The domains crystallized in this study are highlighted by dashed boxes. A sequence conservation map produced by ClustalX2 is shown for the Bub1^CD1 domain. The two sites of phosphorylation (pSer459/pThr461) are highlighted by black arrows. A1: ABBA motif. K1/2: KEN box motifs. MIM: Mad2 interacting motif. CD1: conserved domain 1. BDD: Bub dimerization domain. GLEBS: Gle2-binding-sequence. TPR: Tetratricopeptide repeat. RLK: Arg-Leu-Lys motif. RWD: RING, WD40, DEAD domain.

B  Three views of the crystal structure of Mad1^CTD homodimer (dark/light orange) bound to two Bub1^CD1 peptides (purple/blue). The RLK motif of Mad1 is highlighted in yellow. The two phosphorylation sites are shown as sticks.

termed the Mad1^CTD-Bub1^CD1 complex (Fig 2B) (PDB: 7B1F). This is the first reported structure of the Bub1 CD1 domain and of the Mad1^CTD-Bub1^CD1 complex. The Bub1^CD1 peptide sequence (residues 455–479) used in this study is shown in Fig 2A, with the two sites of phosphorylation (Ser459 and Thr461) indicated. The highest resolution crystal structure, belonging to space group $P2_12_12_1$, diffracted to 1.75 Å, with one Mad1^CTD-Bub1^CD1 complex per asymmetric unit. However, due to strong anisotropic diffraction, the actual resolution is closer to 2.1 Å. A summary of the data collection and refinement statistics is given in Appendix Table S1.

In the crystal structure, two Bub1^CD1 peptides are bound to the Mad1^CTD homodimer (Fig 2B). The structure of Mad1^CTD when

bound to Bub1^CD1 is similar overall to the previously published apo Mad1^CTD crystal structure (PDB: 4DZO) (Kim et al, 2012). Mad1^CTD is comprised of an N-terminal elongated coiled-coil, followed by a globular head domain featuring a conserved RWD (RING, WD40, DEAD) domain (Nameki et al, 2004; Petrovic et al, 2014). Each monomer contains a long N-terminal α-helix (α1) forming the stem, followed by an antiparallel β-sheet of four β-strands (β1–4), a short helix (α2) and two C-terminal helices (α3/α4).

Bub1^CD1 is largely a single α-helix, consisting of almost four turns (Fig 2B). We were able to confidently build 21 of the 26 residues of each Bub1^CD1 peptide, with the first three and last two residues not visible in the electron density map. The six N-terminal residues

including the first phosphorylation site pSer459 are disordered, and extend away from the Bub1$^{CD1}$-Mad1$^{CTD}$ interface (Fig 2B: middle). Unexpectedly, the peptide binding interface lies diagonally across the Mad1$^{CTD}$ coiled-coil, making contacts with both Mad1$^{CTD}$ monomers (Fig 2B: left and right panels). The start of the CD1 helix begins with the phosphothreonine residue (pThr461), contacting the coiled-coil of one monomer at the conserved Mad1 RLK motif, whereas its C-terminus contacts the opposite Mad1 subunit at the top of the coiled-coil and its adjoining head domain β-sheet (Fig 3A–C). The three C-terminal residues of the peptide are disordered (Fig 3A).

Examination of the phosphorylated and unphosphorylated Bub1$^{CD1}$ peptides using $^1$H 1D NMR revealed narrow dispersion of their amide peaks (Appendix Fig S1A), indicating that both peptides are unstructured when not bound to Mad1. Previous circular dichroism experiments also suggested that the unbound Bub1$^{CD1}$ peptide is unstructured but might have helical propensity that is increased with Thr461 phosphorylation (Zhang *et al*, 2017). This indicates that Mad1 binding induces the Bub1$^{CD1}$ domain to adopt a helical conformation, a mechanism which is likely important for the specificity of the Mad1$^{CTD}$-Bub1$^{CD1}$ interaction.

In the complex, Bub1$^{CD1}$ and Mad1$^{CTD}$ interact in a parallel orientation (Fig 2B). This together with the extensive contacts of the C-terminus of the peptide with the head domain of Mad1$^{CTD}$ has implications for the architecture of the Mad1-Mad2-Bub1-Cdc20 assembly at kinetochores. Phosphorylation of Mad1$^{CTD}$ by Mps1, specifically at Thr716, two residues before the C-terminus, has been proposed to promote an interaction with a basic patch on the N-terminal tail of Cdc20 (Ji *et al*, 2017b) (Fig 1A and B). C-terminal to the Bub1 CD1 site is the ABBA and KEN1 motifs (Fig 2A), and although the exact role these motifs play in Cdc20 kinetochore recruitment and MCC formation is still debated, our results suggest that if this model is correct, then the Bub1 CD1 interactions with the head domain of Mad1$^{CTD}$ could have an effect on the Mad1-Cdc20 interaction, particularly as this Mad1-Cdc20 interaction was found to be weak (Vleugel *et al*, 2015; Di Fiore *et al*, 2016; Ji *et al*, 2017b). Together, these multiple interactions of both Bub1 and Mad1 to Cdc20 might be required to properly reposition Cdc20 for binding to the newly converted closed-Mad2, a process which is required for efficient MCC assembly (Luo *et al*, 2002). Additionally, secondary structure prediction of Bub1 residues 448-553 suggests the existence of two α-helices between the CD1 and the ABBA and KEN1 motifs (Buchan & Jones, 2019) (Appendix Fig S2A). Therefore, an appealing idea is that additional contacts between Bub1 and the head domain of Mad1$^{CTD}$ might exist that could be important for proper formation of the Bub1-Mad1-Cdc20 complex.

### The Bub1-Mad1 interaction is mediated by pThr461

Formation of the Bub1-Mad1 complex at kinetochores is dependent on phosphorylation of Bub1, a targeting mechanism conserved from yeast to humans (Heinrich *et al*, 2014; London & Biggins, 2014a; Silió *et al*, 2015). A multiple sequence alignment of the Bub1$^{CD1}$ domain is shown in Fig 2A. In *S. cerevisiae*, phosphorylation of Bub1 at Thr453 and Thr455 by Mps1 promotes an interaction with Mad1$^{CTD}$, with pThr455 alone being sufficient for this interaction (London & Biggins, 2014a; Ji *et al*, 2017b). In humans, the

equivalent phosphorylation sites, Ser459 and Thr461, are also required (Daub *et al*, 2008; Asghar *et al*, 2015; Ji *et al*, 2017b). However, Ser459 is first phosphorylated by Cdk1 that then primes Thr461 phosphorylation by Mps1 (Ji *et al*, 2017b; Qian *et al*, 2017). Similar to yeast, pThr461 alone is sufficient to promote a strong Bub1-Mad1 interaction, whereas phosphorylation of Ser459 is not (Ji *et al*, 2017b; Zhang *et al*, 2017). However, a S459A mutation inactivates Bub1-mediated Mad1 kinetochore recruitment, confirming that this priming mechanism is critical for Bub1$^{CD1}$ functionality (Zhang *et al*, 2017).

Our Mad1$^{CTD}$-Bub1$^{CD1}$ structure is consistent with these findings. Specifically, the phosphate group of pThr461 forms a direct interaction with Mad1 through a charged hydrogen bond with Arg617 of the conserved Mad1 RLK motif (Fig 3C). Additional contacts of the phosphothreonine with Bub1 His463 result in a change in orientation of its imidazole sidechain that enables hydrogen bonding with Ser610 of Mad1. Our structure further reveals that pSer459 does not contact Mad1; instead, it is part of the disordered N-terminal tail of Bub1$^{CD1}$ (Figs 2B and 3A). This further supports the idea that pThr461 is key to promoting the Bub1-Mad1 interaction, whereas Cdk1 phosphorylation of Ser459 primes Mps1 phosphorylation at Thr461.

To further validate the high-resolution structure of Bub1$^{CD1}$ bound to Mad1$^{CTD}$, we performed a phosphorus single-wavelength anomalous dispersion (SAD) experiment (Appendix Table S2). This clearly showed anomalous density for the phosphothreonine at each site built in our structure, confirming the positioning of the phosphate groups (Appendix Fig S3A and B). In contrast, there were no clear signals for the phosphoserine, further suggesting it does not contribute directly to the Bub1$^{CD1}$-Mad1$^{CTD}$ interaction.

We then investigated the interaction of Bub1$^{CD1}$-Mad1$^{CTD}$ using isothermal calorimetry (ITC). We obtained a similar $K_D$ to previously reported binding studies using similar constructs (Ji *et al*, 2017b; Zhang *et al*, 2017), with the doubly phosphorylated pSer459-pThr461 peptide binding to Mad1$^{CTD}$ with a moderate affinity of $2.7 \pm 1.2$ µM (Fig 4A). Both previous studies found a significant reduction in binding for a singly phosphorylated pThr461 peptide. In contrast, we found that the singly phosphorylated peptide bound with a $K_D$ of $3.3 \pm 0.3$ µM (Fig 4B; Appendix Fig S4A), essentially identical to the doubly phosphorylated peptide and consistent with our crystal structure. Our studies therefore suggest that pSer459 does not directly affect the Bub1-Mad1 interaction and supports the idea that pSer459 is predominantly required for priming Thr461 phosphorylation.

Using ITC, we observed no binding of the non-phosphorylated Bub1$^{CD1}$ peptide to Mad1$^{CTD}$ (Fig 4B; Appendix Fig S4B). Additionally, as assessed using NMR spectroscopy, titration of the non-phosphorylated peptide into $^{15}$N-labelled Mad1$^{CTD}$ also failed to reveal Bub1-Mad1 interactions (Appendix Fig S1B). These results further confirm the essential role of the pThr461 phosphorylation site in generating the human Bub1-Mad1 complex.

A striking feature of the pThr461 residue is its position at the start of the CD1 helix. Here, the phosphate group caps the N-terminus of the helix, stabilizing the positively charged helix dipole with its bulky negative charge (Fig 3A and C). The stabilization of α-helices through compensation of the α-helix macro-dipole by capping residues is widely recognized (Wada, 1976; Hol *et al*, 1978; Chakrabartty *et al*, 1993). Our studies reveal how this is regulated by

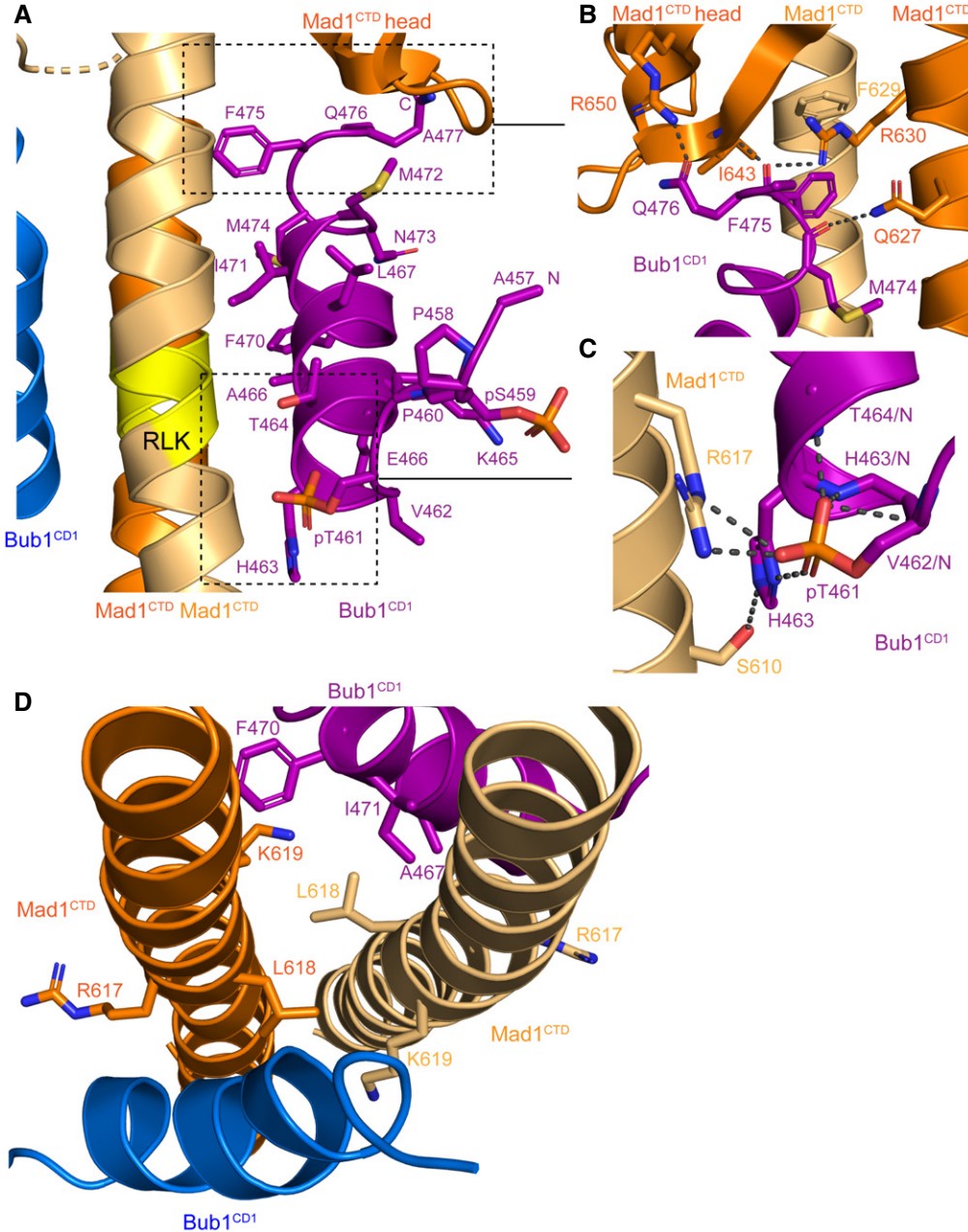

**Figure 3. Molecular interactions of the Mad1$^{CTD}$-Bub1$^{CD1}$ complex.**

A   The extensive largely hydrophobic interface of the Mad1$^{CTD}$-Bub1$^{CD1}$ interaction is highlighted with the higher occupancy CD1 peptide (purple). The RLK motif of Mad1 is emphasized in yellow.

B   Close-up view of Bub1$^{CD1}$ interactions with the head domain of Mad1$^{CTD}$. Hydrogen bonding interactions within 3.5 Å are highlighted by black dashes.

C   Close-up view of the Mad1 Arg617 and Bub1 pThr461 interaction. An additional contact occurs between the phosphate of pThr461 and Bub1 His463 which then forms a hydrogen bond with Mad1 Ser610. Additional stabilizing hydrogen bonding occurs between the pThr461 phosphate and the amide nitrogen atoms of Val462, His463 and Thr464. Hydrogen bonding interactions within 3.5 Å are highlighted by black dashes.

D   Top view of the conserved RLK motifs of the Mad1 homodimer which are shown as sticks. The sidechains of hydrophobic residues near the RLK site at the surface of Bub1$^{CD1}$ are shown as purple sticks which form a hydrophobic pocket.

phosphorylation and is reminiscent of α-helix stabilization by phosphorylation of Ser46 in the bacterial protein HPr (Pullen *et al*, 1995; Thapar *et al*, 1996). Secondary structure prediction suggests that the helix of unphosphorylated CD1 would start from Thr464, further

hinting that pThr461 helps to stabilize an extended helix (Appendix Fig S2A). Using ITC, we found that an R617A mutant of Mad1 severely weakens (3 mM $K_D$), but does not completely abolish, interactions to Bub1$^{CD1}$ (Fig 4B and C, Appendix Fig S5A). This

contrasts with the complete absence of interactions between Mad1[CTD] and the unphosphorylated Bub1[CD1] peptide (Fig 4B; Appendix Figs S1A and S4B). Thus, the phosphate group of pThr461 promotes interactions to Mad1 through direct contacts to Arg617 as suggested by the crystal structure and by promoting conformational changes in the Bub1 CD1 motif. Together, this explains how phosphorylation by Mps1 creates a highly specific interaction between Bub1 and Mad1 and allows for finely tuned regulation of mitotic checkpoint activation.

Phosphorylation of Thr464 within Bub1[CD1], three residues C-terminal to the Thr461 phosphorylation site (Fig 3A), has been reported (Ji et al, 2017b). In vitro analysis suggested that a triply phosphorylate pSer459-pThr461-pThr464 peptide has a lower affinity for Mad1 than the doubly phosphorylated peptide (Ji et al, 2017b). This is explained by our structure because a bulky phosphate at Thr464 would clash with the Bub1[CD1] binding interface of Mad1. Therefore, phosphorylation at Thr464 could provide a means to negatively regulate the Bub1-Mad1 interaction in response to checkpoint signals.

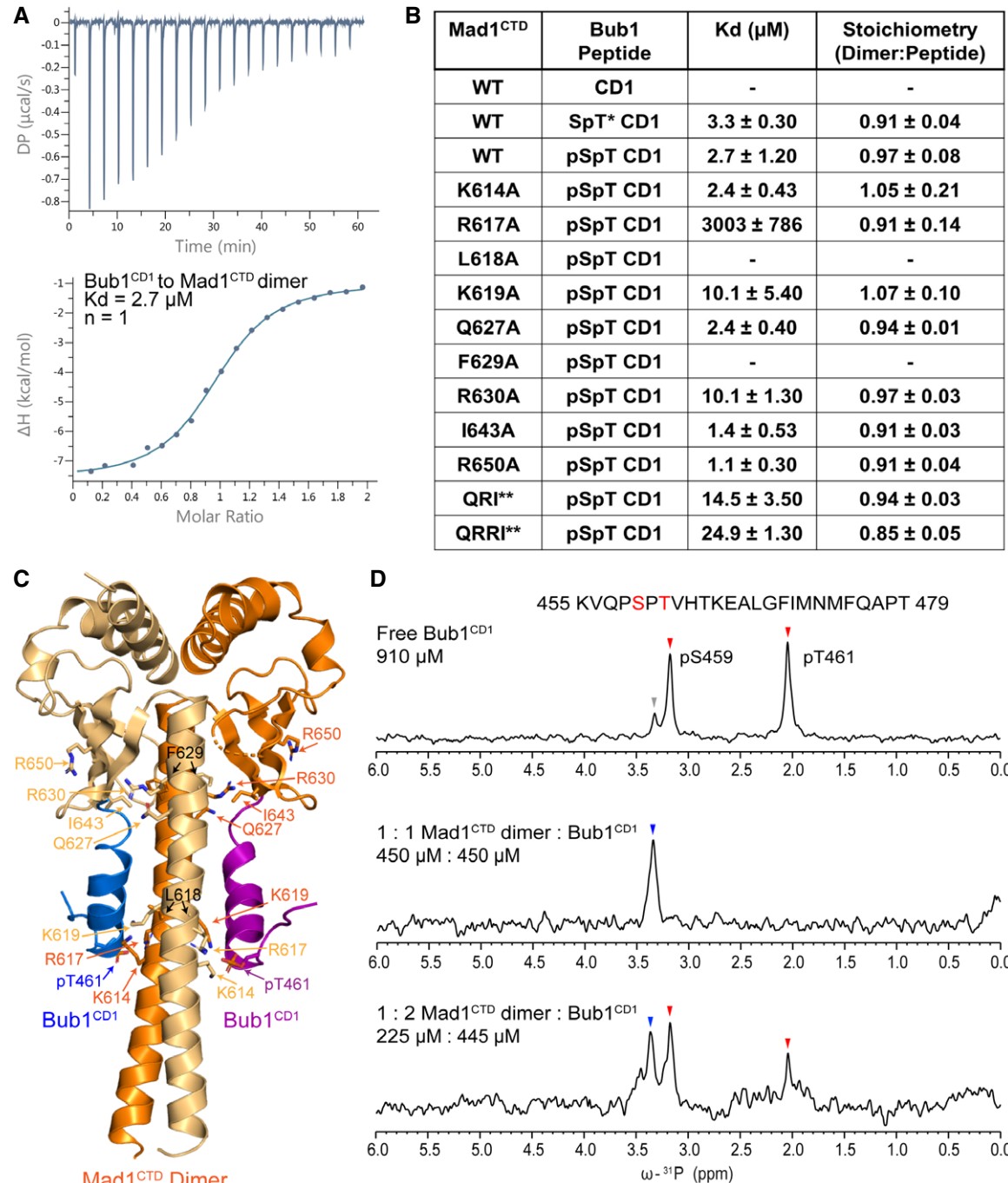

**Figure 4.**

**Figure 4. In solution, only one Bub1$^{CD1}$ binds to the Mad1$^{CTD}$ dimer.**

A  Isothermal calorimetry (ITC) of Mad1$^{CTD}$ binding to doubly phosphorylated Bub1$^{CD1}$. Bub1$^{CD1}$ was injected into Mad1$^{CTD}$ in 19 injections of 2 µl, revealing a dissociation constant (kD) of approximately 2.7 µM and a stoichiometry of 1:1 Mad1$^{CTD}$ dimer to Bub1$^{CD1}$ peptide.
B  Summary of all ITC experiments performed in this study. The K$_D$ and stoichiometry (n) values were obtained by averaging at least three experiments. The reported error values are calculated standard deviations. The mutations of Mad1$^{CTD}$ are highlighted in the Bub1$^{CD1}$-Mad1$^{CTD}$ crystal structure in (C). Mutants which do not bind are marked with a dash. Raw data for each ITC reaction are shown in Appendix Figs S5, S6 and S8. SpT* CD1 peptide is a peptide which is phosphorylated at Thr461 and not at Ser459. QRI** and QRRI** are triple and quadruple mutants of Mad1 which contact the C-terminus of Bub1$^{CD1}$. The QRI** triple mutant contains Q627A, R650A and I643A, while the QRRI** quadruple mutant additionally contains R630A.
C  The crystal structure of Bub1$^{CD1}$-Mad1$^{CTD}$ with the Mad1 residues which were mutated in our ITC experiments shown as sticks. Residues from both monomers are shown. The lower occupancy peptide is shown as blue and the higher occupancy peptide as purple.
D  $^{31}$P 1D spectra showing phosphorylated Bub1$^{CD1}$ peptide titrated with an increasing concentration of Mad1$^{CTD}$ dimer. The peptide sequence is shown above with the two phosphorylated residues highlighted in red. Peaks corresponding to pSer459 and pThr461 are marked with red arrows. The major pSer459 peak represents pSer459 in a trans Ser-Pro bond, and the minor pSer459 peak (marked grey) represents pSer459 in a cis Ser-Pro bond. In the 1:1 molar ratio of Mad1$^{CTD}$ dimer to Bub1$^{CD1}$, the pThr461 peak is significantly line broadened and the pSer459 peak (marked blue) is perturbed. In the 1:2 molar ratio of Mad1$^{CTD}$ dimer to Bub1$^{CD1}$, in addition to signal for the bound pSer459, there is reappearance of the original free position of pS459 and unbound pThr461 supporting the presence of unbound peptide.

## The role of the Mad1 RLK motif

The RLK motif of Mad1 is essential for Mad1 SAC-dependent kinetochore targeting and the site of direct contact with Bub1 (Brady & Hardwick, 2000; Kim *et al*, 2012; Heinrich *et al*, 2014). Except in a few species, such as *C. elegans* and *X. laevis*, the RLK motif is highly conserved (Appendix Fig S2B). However, until now the exact role the RLK motif plays in Bub1 binding has been unclear. Guided by our crystal structure, we investigated the role of individual residues of the RLK motif.

As previously discussed, the interface between Mad1 and Bub1 is mediated by a direct interaction between Mad1 Arg617 of the RLK motif and Bub1 pThr461 (Fig 3C). An alanine mutation of the Arg617 residue nearly abolishes Bub1$^{CD1}$ binding by decreasing the affinity 1,000-fold to the mM range, which confirms the crucial role of this interaction (Fig 4B and C, Appendix Fig S5A).

A L618A mutation of the RLK motif abolished Bub1 binding (Figs 3D and 4B and C; Appendix Fig S5B). This mutant had poor expression compared with wild-type Mad1$^{CTD}$, aggregated easily, and eluted more broadly from a size exclusion column (Fig EV1A). Leu618 lies between the coiled-coil dimerization interface of the Mad1 homodimer (Fig 3D); however, SEC-MALS verified that the purified mutant was a dimer (Fig EV1A). We therefore speculate that this mutant might perturb the Mad1 dimerization interface in the region of the RLK motif and possibly disrupt the Arg617 interaction with Bub1 pThr461. Furthermore, Leu618 contributes to the hydrophobic interface of Mad1$^{CTD}$-Bub1$^{CD1}$ by forming a hydrophobic pocket with neighbouring Phe470 and Ile471 residues on the outside of the Bub1$^{CD1}$ helix (Fig 3D). A L618A mutant would therefore weaken these hydrophobic interactions. Consequently, the lack of detectable interaction between Mad1$^{CTD}$ L618A and Bub1$^{CD1}$ likely results from mis-folding of the Mad1$^{CTD}$ dimer and disrupted Mad1$^{CTD}$-Bub1$^{CD1}$ contacts.

A previously reported kinetochore localization study revealed a key role for Lys619 of the RLK motif because a single Mad1 mutation at this site was defective in kinetochore targeting (Kim *et al*, 2012). To our knowledge, no other study has explored the function of Mad1 Lys619. Surprisingly, in our structure Lys619 of either Mad1 subunit does not form extensive contacts with Bub1 (Figs 3D and 4C). As for the apo structure of Mad1$^{CTD}$, the Lys619 sidechain reaches across the Mad1$^{CTD}$ dimer to the opposite coiled-coil and forms only weak contacts with the opposite Bub1$^{CD1}$ peptide, likely through a π–cation interaction between Mad1 Lys619 and Bub1 Phe470 (Fig 3D). Substituting Ala for Lys619 moderately reduced the Bub1$^{CD1}$-Mad1$^{CTD}$ affinity from 3 to 10 µM (Fig 4B and C; Appendix Fig S5C). These results suggest that the lysine of the RLK motif, although important, is not essential for the Bub1-Mad1 interaction *in vitro* and might point to an additional unknown role of Lys619 in Mad1 kinetochore recruitment.

## Bub1$^{CD1}$ interactions with the Mad1$^{CTD}$ head domain

An important feature of the Bub1$^{CD1}$-Mad1$^{CTD}$ complex is that the C-terminus of each Bub1$^{CD1}$ peptide contacts the top of the coiled-coil and head domain of Mad1 of the opposite subunit to which the phosphorylated threonine interacts (Fig 3A and B). We created individual or combined alanine substitutions for several of these residues (Q627, F629, R630, I643, R650; highlighted in Figs 3B and 4C), and tested to what extent these contacts contribute to the Bub1$^{CD1}$-Mad1$^{CTD}$ interaction using ITC (Fig 4B; Appendix Fig S6A–G).

The only individual mutations which had a significant effect on Bub1$^{CD1}$ binding were F629A and R630A (Fig 4B and C; Appendix Fig S6B and C). Phe629 is a buried hydrophobic residue which contacts Phe629 of the opposite subunit and therefore contributes to the hydrophobic dimerization interface of Mad1$^{CTD}$. Phe629 also contributes to the hydrophobic interface of the Bub1$^{CD1}$-Mad1$^{CTD}$ complex by stacking with Bub1 Phe475 (Fig 3B). A F629A mutation completely abolished Bub1$^{CD1}$ binding (Fig 4B; Appendix Fig S6B). However, we suspect this could be linked to disruption of the Mad1 conformation since although dimeric (Fig EV1A), the F629A mutant was poorly expressed and had a propensity to aggregate. A Mad1 R630A mutant significantly reduced Bub1$^{CD1}$-Mad1$^{CTD}$ binding from 2.7 to 10 µM, suggesting an important role in mediating Bub1$^{CD1}$-Mad1$^{CTD}$ interactions (Fig 4B and C; Appendix Fig S6C). The significant effects of both the F629A and R630A mutants agree with previous reports in which both mutants were defective in kinetochore targeting (Ji *et al*, 2017b).

Although individual alanine substitutions of Gln627, Ile643 and Arg650 had no noticeable effect on Bub1$^{CD1}$ binding (Fig 4B and C; Appendix Fig S6A, D, and E), a triple alanine mutant (termed QRI*) reduced affinity fivefold (to 14.5 µM) indicating that in combination they contribute to the Bub1$^{CD1}$-Mad1$^{CTD}$ interaction (Fig 4B;

Appendix Fig S6F). A quadruple mutant (termed QRRI*), which additionally includes the R630A mutation, further reduced the binding affinity to 25 μM, confirming the significant role of R630A in the Bub1$^{CD1}$-Mad1$^{CTD}$ interaction (Fig 4B; Appendix Fig S6G).

Mutation of the hydrophobic MFQ sequence (residues 474-476) to RRK, located near the C-terminus of Bub1$^{CD1}$, abolished Bub1$^{CD1}$ binding to Mad1$^{CTD}$ (Zhang *et al*, 2017), a finding that confirms the importance of these residues in mediating Bub1$^{CD1}$ interactions with the head domain of Mad1, as revealed by our structure (Fig 3B). Together, these results suggest that the interactions between the C-terminus of Bub1$^{CD1}$ and the Mad1$^{CTD}$ head are important, although contribute less binding energy than the pThr461:Arg617 interaction. In relation to the model of the Bub1:Mad1:C-Mad2:Cdc20 complex at kinetochores shown in Fig 1, it is possible that the contacts of the C-terminus of the Bub1$^{CD1}$ peptide with the head domain of Mad1 are primarily important for positioning Cdc20 in close proximity to the Mad1:C-Mad2 complex.

## In solution, only one Bub1$^{CD1}$ molecule binds the Mad1$^{CTD}$ homodimer

When investigating the Mad1-Bub1 interaction using ITC, we were surprised that Bub1$^{CD1}$ bound to Mad1$^{CTD}$ with a clear stoichiometry of only one peptide per Mad1 dimer (Fig 4A and B). This is not consistent with the two Bub1$^{CD1}$ peptides bound to the Mad1$^{CTD}$ dimer in our crystal structure. This one Bub1$^{CD1}$ peptide to one Mad1$^{CTD}$ dimer stoichiometry was reproducible in eight ITC experiment repeats with wild-type Mad1$^{CTD}$ and Bub1$^{CD1}$ and in all ITC experiments with Mad1 and Bub1 mutants where at least three replicates were performed. The same $K_D$ and stoichiometry were obtained when using a peptide incorporating an N-terminal tryptophan residue to ensure accuracy in peptide concentration measurements. Reversing the titrating species by loading peptide into the calorimeter cell and titrating Mad1$^{CTD}$ resulted in the same stoichiometry (Appendix Fig S4C). Additionally, when we tested Bub1$^{CD1}$ binding to longer forms of Mad1 (485–718 and 420–718) in complex with C-Mad2, which forms a Mad1:C-Mad2 tetramer, the stoichiometry was still conserved, although there was a slight decrease in the binding affinity for the tetrameric complexes (Appendix Fig S4D and E). This indicates that at least for the specific binding of the Bub1$^{CD1}$ domain to the Mad1 RLK site, this 1:1 ratio of Bub1$^{CD1}$ to Mad1$^{CTD}$ dimer stoichiometry is likely to be conserved with full-length proteins.

We used NMR spectroscopy to further analyse the stoichiometry of the Bub1$^{CD1}$-Mad1$^{CTD}$ interaction in solution to address the discrepancy between our ITC data and crystal structure. The unique $^{31}$P phosphate 1D NMR signals observed for pSer459 and pThr461 in the phosphorylated Bub1$^{CD1}$ peptide enabled us to probe its interaction with Mad1$^{CTD}$ (Fig 4D). pSer459 and pThr461 are part of a Ser-Pro-Thr sequence and the $^{31}$P NMR spectrum yielded not only two distinct signals for the two phosphorylated residues but also a splitting of the—subsequently confirmed—pSer459 signal as a result of the cis-trans proline isomerization of the Ser-Pro amide bond. At a 1:1 ratio of Bub1$^{CD1}$ to Mad1$^{CTD}$ dimer, the free pThr461 signal disappeared indicating that pThr461 is a key residue for the Bub1$^{CD1}$-Mad1$^{CTD}$ interaction. The pSer459 signal retained all the characteristics of the free species apart from a small shift, suggesting that pSer459 remains highly flexible and its chemical environment is only

marginally affected by the now bound Bub1$^{CD1}$ peptide. These results further confirm the interactions seen in our crystal structure in which pThr461 mediates contacts to Mad1$^{CTD}$ whereas pSer459 does not.

Increasing the Bub1$^{CD1}$ concentration to a 2:1 ratio relative to the Mad1$^{CTD}$ dimer resulted in a complex spectrum with multiple signals for both phosphorylated residues (Fig 4D). Striking and easy to interpret were the two signals observed for pSer459 at the bound and the original free position, indicating that the spectrum was the overlay of free and bound phosphorylated Bub1$^{CD1}$ peptide at equivalent stoichiometries. The reappearance of a signal at the chemical shift position for unbound pThr461 further supports the presence of unbound peptide. Taken together, our results provide strong evidence that the preferred binding mode is in a 1:1 ratio of Bub1$^{CD1}$ to the Mad1$^{CTD}$ dimer. Increasing the Bub1$^{CD1}$ concentration did not result in a second binding event.

We also investigated the stoichiometry of the Mad1$^{CTD}$-Bub1$^{CD1}$ complex using analytical ultracentrifugation (AUC) sedimentation equilibrium (Fig EV1B). Mad1$^{CTD}$ at 20 μM was incubated with either 20 or 40 μM Bub1$^{CD1}$ peptide. In both cases, the calculated mass was close to the expected mass of a Mad1$^{CTD}$ dimer bound to a single Bub1$^{CD1}$ peptide, further confirming the 1:1 stoichiometry of the Mad1$^{CTD}$ dimer to the Bub1$^{CD1}$ peptide observed by NMR and ITC.

Our ITC and sedimentation equilibrium experiments to measure the stoichiometry of the Mad1$^{CTD}$-Bub1$^{CD1}$ interaction were performed with concentrations of Mad1 at 100 μM or lower, and titrating Bub1 until a twofold-fourfold molar excess was obtained. However, our $^{31}$P NMR experiments were conducted with around 0.25 to 0.5 mM of each, and still only one binding event was detected. Our ability to test binding at higher concentration was limited by the poor solubility of the peptide above 1.5 mM without the addition of DMSO. DMSO interfered with ITC enthalpy changes and the NMR signal from DMSO obscured the protein peaks. In contrast, our Mad1$^{CTD}$-Bub1$^{CD1}$ crystal structure was obtained through co-crystallization of Mad1$^{CTD}$ at a final concentration of 0.35 mM and Bub1$^{CD1}$ at 2.5 mM in 5% DMSO. The use of millimolar concentrations of peptide, and the presence of DMSO and isopropanol, required for Mad1$^{CTD}$-Bub1$^{CD1}$ crystallization, may explain the association of a second peptide in the crystallized complex. Notably, the concentrations of Mad1 and Bub1 used for crystallization greatly exceed their sub-micromolar concentrations at the kinetochore (Faesen *et al*, 2017). Additionally, the absence of crystal contacts involving the Bub1$^{CD1}$ peptide suggests that the 2:2 stoichiometry of the Mad1$^{CTD}$-Bub1$^{CD1}$ complex is not a crystallization artefact.

In our crystal structure, where Bub1$^{CD1}$ and Mad1$^{CTD}$ are co-crystallized at concentrations over 100 times the $K_D$ and with a sevenfold molar excess of peptide, we see clear differential occupancy of the peptides in the electron density map and the 2Fo-Fc omit map (Figs EV2A and C, and D). This suggests that in the crystal the two peptides display differing affinities for Mad1$^{CTD}$, despite the interactions between Mad1$^{CTD}$ and the two peptides being largely conserved (compare Fig 3A–C with Fig EV2C, and E–G). Crystallization of Mad1$^{CTD}$-Bub1$^{CD1}$ with a twofold higher peptide concentration of 5 mM led to a lower resolution structure most likely due to the twofold increase in DMSO. However, the differential occupancy, although reduced, was still observed in this structure (Fig EV2B).

Lowering the peptide and/or DMSO concentration resulted in either an increase in the differential occupancy of the peptides or crystallization of apo Mad1$^{CTD}$. We assume one reason for this is

that in the apo Mad1$^{CTD}$ structure (P6 crystals), the close packing of Mad1$^{CTD}$ dimers blocks Bub1$^{CD1}$ peptide binding. Attempts to soak the Bub1$^{CD1}$ peptide at high concentration into these apo Mad1$^{CTD}$ crystals were unsuccessful. We assume that at lower concentrations of peptide, the tight P6 packing of the apo Mad1$^{CTD}$ is favoured.

We note that fluorescent quantification of budding yeast Bub1 and Mad1 suggested that kinetochores recruit two Mad1:C-Mad2 complexes per Bub1-Bub3 (Aravamudhan *et al*, 2016). Cellular concentration studies in mammalian cell culture systems indicated 100 nM of Bub1 and 20 nM of Mad1 (Howell *et al*, 2004; Luo *et al*, 2004; Shah *et al*, 2004). We therefore cannot exclude the possibility that the Bub1-Mad1 complex presents a different stoichiometry *in vivo;* however, our results strongly point towards a mechanism by which only one Bub1 molecule binds to the Mad1 homodimer. This has interesting implications for how MCC is catalytically assembled onto the outer kinetochore. It has been suggested that Mad1$^{CTD}$ might fold back onto the Mad1:C-Mad2 core to promote template conversion of Mad2 from an open to a closed state (De Antoni *et al*, 2005; Mapelli *et al*, 2007). If this fold-over model is correct, it might imply that only one site is either available or favourable for Bub1 binding. Our model of the Bub1-Mad1-Cdc20 interaction at kinetochores (Fig 1A and B) suggests that only one Cdc20 molecule can be accommodated at the head domain of Mad1 proximal to the Thr716 phosphorylation site, especially if one side of the Mad1 head was making contact with the Mad1:C-Mad2 core. Therefore, this could be another reason only one Bub1 binds.

Altogether, these results led us to examine the Mad1$^{CTD}$-Bub1$^{CD1}$ crystal structure in more detail, as well as to further investigate the Bub1-Mad1 interaction in solution by NMR, to explain why one peptide would have preferential binding to the Mad1 homodimer.

## The Mad1$^{CTD}$ homodimer is asymmetric

The structure of the previously crystallized apo Mad1$^{CTD}$ homodimer is markedly asymmetric (Fig EV3A and B) (Kim *et al*, 2012). In particular, there is a large asymmetric curvature in the coiled-coil stem. One coiled-coil α-helix (apoB) is relatively straight whereas its counterpart (apoA) is significantly bent. Interestingly, this hinge bending occurs at the RLK motif (Fig EV3C). Aligning the RLK motif of both subunits of the apo homodimer shows that the head domain is rotated inwards towards the more bent α-helix (Fig EV3C). This is due to a change in the angle at which both helices of the coiled-coil are curved with respect to the head domain, rather than a change in the confirmation of the head domain itself (Fig EV3D). This asymmetry of apo Mad1 is also reflected in the relative B-factors, with the more bent protomer exhibiting higher flexibility (Fig EV3E).

Bub1$^{CD1}$-bound Mad1$^{CTD}$ also displays a similar type of asymmetry but to a reduced extent (Fig 5A). One Mad1$^{CTD}$ subunit is slightly more curved than the other, but there is an overall straightening of the more bent subunit compared with the apo structure (Fig 5A). However, as for the apo structure, both subunits of the coiled-coil are strikingly arched to one side (Fig 5B). Significantly, the side of the coiled-coil which angles inwards comprises the binding site of the higher occupancy peptide, whereas the lower, partial occupancy peptide binds to the outside of the bend (Fig 5A and B). This curvature of the coiled-coil results in stronger engagement of both the Mad1 head domain and the hydrophobic coiled-coil with the higher occupancy peptide (Fig 5A and C).

Comparing how each side of the Mad1 head domain engages with the adjacent peptide shows that there are additional hydrogen bond contacts with the C-terminus of the higher occupancy peptide compared with the lower occupancy peptide (Fig 5C). Most noticeable are contacts between Bub1 Gln476 and Mad1 Arg650, as well as Bub1 Met474 and Mad1 Gln627 in the higher occupancy peptide (Fig 5C; purple) which do not exist in the lower occupancy peptide (Fig 5C; blue). However, because our ITC experiments show that neither the Arg650 nor Gln627 residues have a strong influence on the binding of the peptide to Mad1$^{CTD}$, it is unlikely that the differential occupancy is solely the result of how the head domain engages each peptide. More likely, the differential occupancy is a result of the net effect of the peptide binding to the inside of the bent coiled-coil, rather than to the outside of the bend, combined with enhanced contacts to the head domain.

This asymmetry within Bub1$^{CD1}$-Mad1$^{CTD}$ homodimer is also clearly demonstrated by mapping the B-factors onto the structure (Fig 5D). The side of the homodimer (bound_A and head_B) which contacts the higher occupancy peptide (high_CD1) has higher rigidity as does the peptide itself. It is therefore probable that this asymmetry of the Bub1 bound Mad1 homodimer leads to the differential peptide occupancy seen in our X-ray structure.

Crystallographic packing is unlikely the cause of the asymmetry present in the apo Mad1$^{CTD}$ and Bub1$^{CD1}$-bound Mad1$^{CTD}$ structures. Both the apo and Bub1$^{CD1}$-bound states are asymmetric, yet crystallized in different space groups. Furthermore, we obtained several crystal structures of the Mad1$^{CTD}$-Bub1$^{CD1}$ complex using the same crystallization conditions but in different space groups (Fig EV4A–G and Appendix Table S1). In each case, the asymmetry of the homodimer was retained. The asymmetric unit of the monoclinic space group (P2$_1$) comprises two Mad1$^{CTD}$-Bub1$^{CD1}$ complexes (PDB: 7B1H). Both complexes display asymmetry, however with differing degrees (Fig EV4C–F). In the more asymmetric complex, there is a more marked difference in Bub1$^{CD1}$ peptide occupancy, with high occupancy for the peptide bound to the concave side of the Mad1$^{CTD}$ coiled-coil, and very low occupancy for the Bub1$^{CD1}$ peptide bound to the convex side of the Mad1$^{CTD}$ coiled-coil. It is therefore likely that the asymmetry is intrinsic to the Mad1$^{CTD}$ homodimer and contributes to the differential occupancy of Bub1$^{CD1}$ peptides in our crystal structures. Alignment of all four Mad1$^{CTD}$ homodimers from the three different space group structures displays the extent of the flexibility within the Mad1$^{CTD}$ coiled-coil and head domain (Fig EV4G).

To our knowledge, we are the first to comment on this asymmetry in Mad1$^{CTD}$. However, before our study there would have been no obvious significance for this asymmetry, and the asymmetry of apo Mad1$^{CTD}$ could have been easily attributed to crystallization artefacts. Interestingly, the crystal structure of the tetrameric Mad1$^{485–584}$:C-Mad2 complex is also asymmetric (Sironi *et al*, 2002). The N-terminal coiled-coil of Mad1 undergoes a break where a linker region (residues 531–539), directly followed by the Mad1 MIM motif, adopts a different confirmation in each Mad1 protomer, resulting in a significantly asymmetric tetramer. Although the functional significance of this asymmetry was not demonstrated, the authors hypothesized that the asymmetry might be connected to Mad2 conversion at kinetochores. An appealing idea is that the asymmetry of both the Mad1:C-Mad2 tetramer and the flexibly

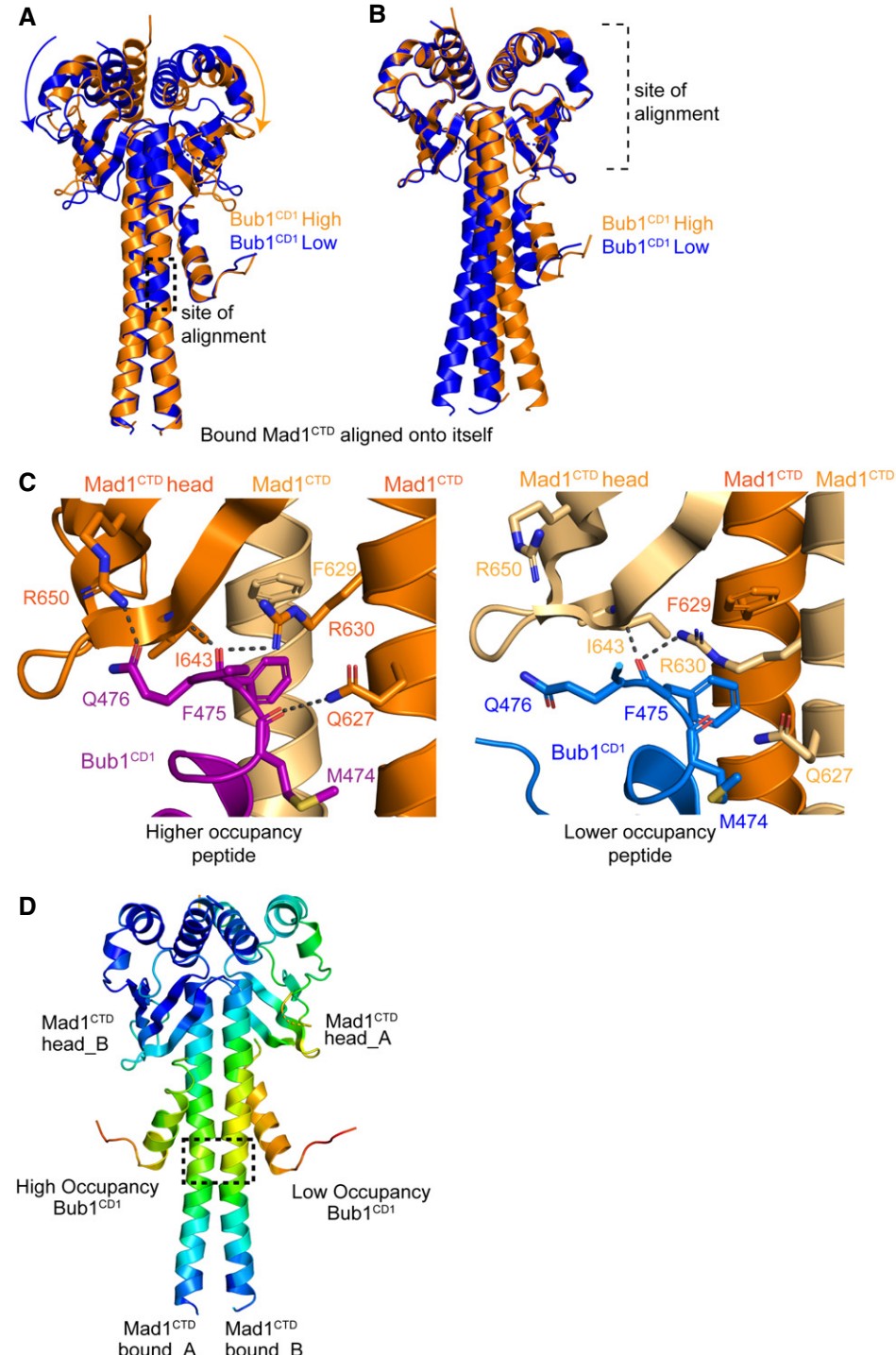

**Figure 5. Bub1^CD1 bound Mad1^CTD homodimer is asymmetric.**

A  Alignment of the RLK motif of opposite subunits of the Mad1^CTD-Bub1^CD1 structure. The site of alignment is highlighted by the dashed box. In one dimer (coloured orange), the higher occupancy peptide is depicted as an orange cartoon. In the other dimer (coloured blue), the lower occupancy peptide is depicted as a blue cartoon. The orange and blue arrows highlight how there is stronger engagement of the head domain with the higher occupancy peptide.

B  Alignment of the head domain of opposite subunits of the Mad1^CTD-Bub1^CD1 structure. In one dimer (coloured orange), the higher occupancy peptide is depicted as an orange cartoon. In the other dimer (coloured blue), the lower occupancy peptide is depicted as a blue cartoon.

C  Comparison of the high and low occupancy Bub1^CD1 peptide contacts with the opposite sides of the Mad1 head domain and top of the coiled-coil. Hydrogen bonds within 3.5 Å are shown as black dashes.

D  Temperature factors mapped onto the Mad1^CTD-Bub1^CD1 crystal structure.

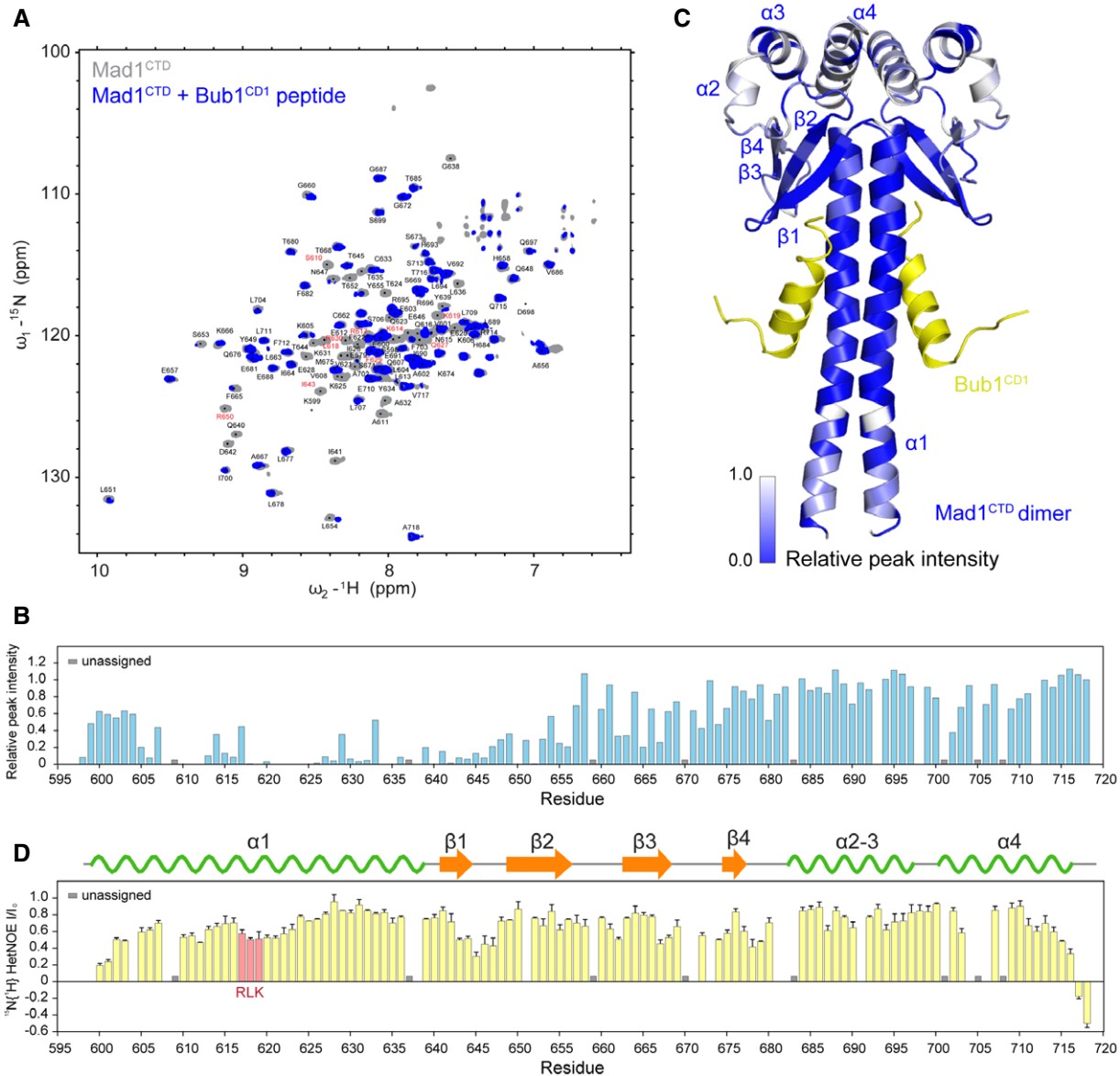

**Figure 6. The Bub1-Mad1 interaction characterized by NMR.**

A   $^1H,^{15}N$-2D HSQC showing $^{15}N$-labelled Mad1$^{CTD}$ with (blue) and without (grey) Bub1 phosphorylated peptide. Peptides were added in excess at 1:2 molar ratio of Mad1$^{CTD}$ dimer to Bub1$^{CD1}$. Assignments of the backbone resonances of Mad1$^{CTD}$ are labelled on the spectra. Close-up of the chemical shift perturbations of the residues in red during Bub1$^{CD1}$ titration, are shown in Appendix Figs S7 and S8.

B   Relative peak intensities (bound/free) of Mad1$^{CTD}$ upon Bub1$^{CD1}$ binding. Peak intensities were normalized to that of the C-terminal residue Ala718.

C   Relative peak intensities of Bub1$^{CD1}$-bound Mad1$^{CTD}$ in B are mapped onto the Bub1$^{CD1}$-Mad1$^{CTD}$ crystal structure. Residues are coloured in a scale of blue to grey, where regions with the most significant line broadening are highlighted in blue.

D   $^{15}N\{^1H\}$-heteronuclear NOE values were collected with interleaved on- (I) and off- (I$_0$) resonance and expressed as I/I$_0$. A higher value indicates higher rigidity of the backbone N–H bond. The Mad1 RLK motif is highlighted in red. The error bars are the calculated standard deviations of two technical replicates on the same sample.

tethered Mad1$^{CTD}$ might have a common function in regulating MCC assembly at kinetochores.

## Bub1 binding leads to substantial conformational changes in Mad1$^{CTD}$

NMR spectroscopy was used to gain detailed structural insights into the Bub1$^{CD1}$-Mad1$^{CTD}$ interaction in solution based on an assignment of Mad1$^{CTD}$ backbone resonances. To counteract a reduction in sensitivity associated with the slow overall tumbling of the elongated Mad1$^{CTD}$ dimer, we employed relaxation optimized 3D experiments using a uniformly sidechain deuterated $^{13}C,^{15}N$-labelled sample. Backbone H$_N$, N, C$\alpha$, C$\beta$ resonances for 113 out of 119 non-proline residues were assigned, and their conformation-dependent secondary chemical shifts confirmed secondary structure elements observed in the crystal structure (Fig 6A and D).

For the analysis of residues involved in peptide binding, unlabelled phosphorylated Bub1 peptide was added to $^{15}$N-labelled Mad1$^{CTD}$. We observed a substantial attenuation for resonances corresponding to residues 605 to 655 in the $^{1}$H,$^{15}$N correlation spectra of Mad1$^{CTD}$ (Fig 6A and B; Appendix Figs S7 and S8). Mapping these line broadened residues onto the crystal structure of the Bub1$^{CD1}$-Mad1$^{CTD}$ complex revealed the extent of environmental changes upon peptide binding (Fig 6C). It is important to note that in general, NMR titrations yield either actual changes in the chemical shift positions of residues affected or, as found here, line broadening, especially in situations where binding is in the lower μM regime. Changes indicate the actual binding site but also general conformational rearrangements as a consequence of complex formation. An additional complication arises from the fact that Mad1 as a dimer only displays one set of signals which indicates conformational exchange between the two monomeric species. Attenuated signals correspond to most of the coiled-coil region and two of the β-strands in the head domain (Fig 6C). This indicates that not only the Mad1 RLK motif is involved in Bub1 binding but also large structural elements of Mad1 experience changes in their chemical environment. Although the addition of phosphorylated Bub1$^{CD1}$ peptide to $^{15}$N-labelled Mad1$^{CTD}$ resulted in major signal attenuations at a molar ratio of 1:1 Mad1$^{CTD}$ dimer to Bub1$^{CD1}$ peptide, further addition of peptide did not yield significant additional changes (Fig EV5A and B). This supports the 1:1 Bub1$^{CD1}$ to Mad1$^{CTD}$ dimer stoichiometry observed in our ITC, $^{31}$P NMR and sedimentation equilibrium experiments and argues against a second binding site in solution.

A $^{15}$N{$^{1}$H}-heteronuclear NOE experiment that samples $^{15}$N backbone dynamics on a fast picosecond time scale revealed the RLK motif of free Mad1$^{CTD}$ as the most flexible segment of the coiled-coil region (Fig 6D). This supports the idea that this motif acts as a dynamic hinge resulting in a bend within the apo coiled-coil region and most likely also in the bound structures. Although it is conceivable that complex formation with Bub1$^{CD1}$ will result in dynamic changes, substantial line broadening of the Bub1$^{CD1}$-bound Mad1$^{CTD}$ prevented us from obtaining dynamic data for the complex. Nevertheless, one can speculate that upon interaction with Bub1, the rigidity of the RLK motif increases and this in turn would result in an increase in the overall rigidity of the coiled-coil.

Altogether, our NMR data suggest dynamic changes and local conformational rearrangements within Mad1$^{CTD}$ upon Bub1$^{CD1}$ binding. This is consistent with the multiple contacts observed in the Mad1$^{CTD}$-Bub1$^{CD1}$ structure. In particular, the substantial line broadening observed in the coiled-coil region can be explained by the fact that the peptide is bound diagonally across the Mad1$^{CTD}$ coiled-coil and therefore makes extensive contacts with both subunits. With the effect of Bub1 binding experienced in nearly the entire coiled-coil region, it is conceivable that a conformational rearrangement in the bend of the coiled-coil region of the apo Mad1$^{CTD}$ is a requirement for efficient Bub1$^{CD1}$ binding.

### Mad1 asymmetry likely controls Bub1$^{CD1}$ binding stoichiometry

To assess whether the conformation of the apo Mad1$^{CTD}$ crystal structure is compatible with Bub1$^{CD1}$ binding, we superimposed the apo and Bub1$^{CD1}$-bound Mad1$^{CTD}$ molecules by aligning them on the RLK motif of each subunit (Appendix Fig S9). A total of four alignment combinations are possible. Alignments placing either of the two bound Bub1 peptides on the concave side of the apo structure coiled-coil cause severe clashes (Appendix Fig S9A and C), whereas contacts between either peptide and the Mad1$^{CTD}$ head domain are lost on the convex side, especially for the higher occupancy peptide (Appendix Fig S9B and D). This analysis therefore indicates that Bub1$^{CD1}$ is unable to interact with the apo conformation of Mad1$^{CTD}$ seen in the crystal structure.

A transition of Mad1$^{CTD}$ to a more symmetric state, involving a conformational change centred on the flexible RLK motif, straightening the coiled-coil, allows Bub1$^{CD1}$ binding. Such a conformation, observed in our Mad1$^{CTD}$-Bub1$^{CD1}$ crystal structure, is consistent with the conformational changes occurring to Mad1$^{CTD}$ upon Bub1 binding in solution, as detected by NMR (Fig 6). The higher the occupancy the peptide of the Mad1$^{CTD}$-Bub1$^{CD1}$ crystal structure is indeed the one at the concave side of the coiled-coil forming more contacts to the Mad1$^{CTD}$ head domain, consistent with mutagenesis and ITC data (Appendix Fig S6). Thus, this peptide is likely to bind with higher affinity and to represent the single binding peptide in solution as detected by NMR, ITC and AUC analyses. This model is supported by our Mad1$^{CTD}$-Bub1$^{CD1}$ structures from various space groups where higher occupancy of the second Bub1$^{CD1}$ peptide correlates with a more symmetric Mad1$^{CTD}$ dimer (Fig EV4).

## Discussion

This study provides insights into the mechanism of how the Mad1: C-Mad2 complex is targeted to kinetochores in response to SAC activation, a process regulated by a sequential Mps1-dependent phosphorylation cascade (Fig 1). Cdk1 and Mps1 phosphorylate the Bub1 CD1 domain to create a direct interaction with the C-terminus of Mad1. Our Mad1$^{CTD}$-Bub1$^{CD1}$ crystal structure explains the molecular interactions of this highly specific targeting mechanism. We find that the first Bub1 phosphorylation site, pSer459, does not make direct contact with Mad1, consistent with the suggested mechanism by which Cdk1 phosphorylation of Ser459 primes Mps1 phosphorylation of Thr461 (Ji *et al*, 2017b). We show that pThr461 directly binds to Mad1 Arg617 of the conserved RLK motif, and we suggest that the high specificity of this interaction results from the ability of the phosphate of pThr461 to stabilize the N-terminal α-helix dipole of Bub1$^{CD1}$. Using a variety of biophysical techniques, we determined that only one Bub1$^{CD1}$ peptide binds to the Mad1$^{CTD}$ homodimer in solution. Analysis of apo and bound Mad1$^{CTD}$ crystal structures indicates that the homodimer is intrinsically asymmetric, whereby the Mad1$^{CTD}$ coiled-coil has significant curvature which also causes stronger engagement of the head domain with the peptide bound to the side of the coiled-coil with the concave bend. We suggest that this asymmetry is the reason only one peptide binds in solution. This also explains the differential occupancy of the two peptides bound to the Mad1$^{CTD}$ homodimer in our crystal structure. The use of millimolar concentrations of peptide, and the presence of DMSO and isopropanol, required for Mad1$^{CTD}$-Bub1$^{CD1}$ crystallization, may explain the association of a second peptide in the crystallized complex. Altogether, we propose that the asymmetry of the Mad1$^{CTD}$-Bub1$^{CD1}$ complex is an intrinsic and functional feature that plays an important role in generating the correct juxtaposition of SAC proteins required to catalyse MCC assembly.

# Materials and Methods

### Cloning, expression and purification of Mad1[CTD]

The coding region of Mad1[597–719] was cloned by USER® (NEB) into a modified pRSFDuet-1 vector (71341-3, Sigma-Aldrich) with an N-terminal double Strep His₆-tag, followed by a tobacco etch virus (TEV) protease cleavage site (Demple & Linn, 1982; Bitinaite et al, 1992). Mad1 mutants were generated using the QuikChange™ Lightning Site-Directed Mutagenesis Kit (Agilent), developed by Stratagene Inc. (La Jolla, CA) (Nøhr & Kristiansen, 2003). Mad1 constructs were transformed into Rosetta™ 2 (DE3) Singles™ Competent Cells (71400, Novagen) for expression. Expression was induced with 0.5 mM IPTG and grown overnight at 18°C. Cells were lysed in 25 mM HEPES pH 8.1, 250 mM NaCl, 2 mM DTT, 5% glycerol, 2 mM EDTA supplemented with lysozyme, and Complete™ EDTA-free protease inhibitors (Roche). Proteins were purified over a Strep-Tactin Superflow Plus column (QIAGEN) and cleaved with TEV protease overnight at 4°C. The cleaved Mad1[CTD] was then diluted to 50 mM NaCl, purified over an anion exchange (Resource Q, GE Healthcare) column, followed by size exclusion chromatography using a Superdex 75 Increase column (GE Healthcare). Mad1[CTD] was concentrated to 35 mg/ml in a buffer of 20 mM HEPES pH 7.5, 100 mM NaCl and 1 mM TCEP.

### Peptide synthesis

Bub1[CD1] peptides were ordered from Designer Bioscience UK, at 95% purity. The full peptide sequence used in this study was (W) KVQP{pS}P{pT}VHTKEALGFIMNMFQAPTS. (W) = peptide with an additional N-terminal tryptophan was used to confirm peptide concentration in stoichiometry studies.

### Isotopic labelling of Mad1[CTD]

Isotopically labelled Mad1[CTD] was expressed in M9 minimal media (6 g/l Na₂HPO₄, 3 g/l KH₂PO₄, 0.5 g/l NaCl) supplemented with 1.7 g/l yeast nitrogen base without NH₄Cl and amino acids (Sigma Y1251). 1 g/l ¹⁵NH₄Cl and 4 g/l unlabelled glucose were supplemented for ¹⁵N labelling. For the deuteration of nonlabile sidechain protons, cells were adapted for growth in 10, 44 and 78% deuterated media on agar plates before they were grown in large cultures supplemented with 1 g/l ¹⁵NH₄Cl and 4 g/l ²H¹³C-glucose in 99% D₂O (Sigma). Prior to NMR experiments, isotopically labelled Mad1[CTD] and Bub1[CD1] peptide were dialysed into 25 mM HEPES, pH 7.0, 100 mM NaCl, 1 mM TCEP.

### NMR spectroscopy

¹H-detected experiments were performed on 600 and 800 MHz Avance III spectrometers, both equipped with triple resonance TCI CryoProbes (Bruker). ³¹P 1D NMR spectra were recorded on a 500 MHz Avance II spectrometer equipped with a broadband BBO cryoprobe (Bruker). All spectra were collected in 50 mM HEPES, pH 7.0, 100 mM NaCl, 2 mM TCEP, 0.02% NaN₃ at 298K. Standard ²H decoupled TROSY-based triple resonance experiments were used for backbone resonance assignments: HNCO, HN(CA)CO, HNCA, HN(CO)CA, HNCACB, HN(CO)CACB and HN(COCA)NNH

(Pervushin, 2020). Backbone datasets were collected with 20–25% non-uniform sampling and reconstructed using compressed sensing in MddNMR (Mayzel et al, 2014). Topspin 3.6 (Bruker) was used for processing and NMRFAM-Sparky 1.47 for data analysis (Lee et al, 2015). Backbone assignments were obtained in Mars (Jung & Zweckstetter, 2004). Secondary chemical shifts were calculated from the differences between observed Cα/Cβ chemical shifts and Cα/Cβ chemical shifts for random coils (Kjaergaard & Poulsen, 2011). For binding studies, the relative peak intensities were normalized to the C-terminal residue Ala718 of Mad1[CTD] and expressed as $PI_{bound}/PI_{free}$, with $PI_{bound}$ and $PI_{free}$ being the peak heights of the free and bound forms, respectively. ¹⁵N{¹H}-heteronuclear NOE values are expressed as $I/I_0$ ratio and measured using standard Bruker pulse sequences, with interleaved on- (I) and off-resonance ($I_0$) saturation. ³¹P 1D spectra were recorded using a standard ¹H-decoupled sequence with power-gated decoupling and 30° flip angle.

### Isothermal titration calorimetry

Isothermal titration calorimetry (ITC) was performed using an Auto-iTC200 instrument (Malvern Instruments, Malvern, UK) at 20°C. Mad1[CTD] and Bub1[CD1] peptide mutants were dialysed into 20 mM HEPES pH 7.5, 100 mM NaCl, 1 mM TCEP. For each titration between 30–100 μM of Mad1 was pipetted into the calorimeter cell. The Bub1[CD1] peptides at 0.3–1.5 mM were titrated into the cell consisting of one 0.5 μl injection followed by 19 injections of 2 μl each. The changes in the heat released were integrated over the entire titration and fitted to a single-site binding model using the MicroCal PEAQ-ITC Analysis Software 1.0.0.1258 (Malvern Instruments). Titrations were performed in triplicate. Calculations for stoichiometry were based on the molar concentrations of dimeric Mad1[CTD].

### SEC-MALS

Size exclusion chromatography coupled with multi-angle static light scattering (SEC-MALS) was performed using an Agilent 1200 series LC system with an online Dawn Helios ii system (Wyatt) equipped with a QELS+ module (Wyatt) and an Optilab rEX differential refractive index detector (Wyatt). 100 μl purified protein from 0.5–2.0 mg/ml was auto-injected onto a Superdex 200 Increase 10/300 GL column (GE healthcare) and run at 0.5 ml/min. The molecular masses were analysed with ASTRA 7.3.0.11 (Wyatt). Data were plotted using Prism 8.4.3 (GraphPad Software, Inc).

### Analytical ultracentrifugation

Mad1[CTD] was mixed with Bub1[CD1] peptide to give final concentrations of 20 μM Mad1[CTD] with either 20 or 40 μM Bub1[CD1] in 20 mM HEPES pH 7.5, 100 mM NaCl, 1 mM TCEP. Samples were loaded into 12 mm 6-sector cells, placed in an An50Ti rotor and centrifugated at 10,200, 12,200 and 21,000 rpm at 20°C until equilibrium had been reached using an Optima XL-I analytical ultracentrifuge (Beckman). The data were analysed in SEDPHAT 15.2b (Schuck, 2003). The partial-specific volumes (v-bar) were calculated using Sednterp (Cole et al, 2008). The density and viscosity of the buffer were determined with a DMA 4500M density meter (Anton Parr)

and a AMVn viscometer (Anton Paar). Data were plotted with the program GUSSI (Brautigam, 2015).

## Crystallization

Purified Mad1$^{CTD}$ at 11 mg/ml in 20 mM HEPES pH 7.5, 100 mM NaCl and 1 mM TCEP was mixed for 10 mins with Bub1$^{CD1}$ peptide dissolved into 100% DMSO at 50 mM starting concentration such that the final concentration of Bub1$^{CD1}$ was 5 or 2.5 mM Bub1$^{CD1}$ in either 10 or 5% DMSO, respectively. Sample was trialled in several sparse matrix screens using the sitting-drop vapour diffusion method at 21°C (Jancarik & Kim, 1991; Gorrec, 2016). Crystal hits were obtained in Hampton Research Crystal Screen 1 (HR2-110), 10% isopropanol, 0.1 M Na HEPES pH 7.5, 20% PEG 4000. Crystal growth was optimized using this condition and a drop size of 500 nl of 1:1 protein to reservoir solution. The crystals were harvested within 48 h and flashed cooled in liquid nitrogen using the reservoir solution supplemented with 20% glycerol.

## Crystallographic data collection, structure determination and refinement

Data were collected at beamline I04 at the Diamond Light Source, U.K. Selected data sets were autoprocessed using the XDS pipeline in Xia2 (Kabsch, 2010; Winter, 2010).

Phenix version 1.18.2-3874 was used for structure solution with PHASER-MR molecular replacement against PDB: 4DZO (Kim *et al*, 2012; Liebschner *et al*, 2019). The peptide ligand was manually built in Coot (Emsley *et al*, 2010). Refinement was performed using PHENIX and validation with MolProbity (Williams *et al*, 2018).

## Single-wavelength anomalous dispersion

The crystals were harvested using LithoLoop sample mounts on specialized I23 copper sample assemblies. Data were collected on the long-wavelength beamline I23 at Diamond Light Source at X-ray energy of 4.5 keV using the semi-cylindrical PILATUS 12M (Dectris, CH) (Wagner *et al*, 2016). 4.5 keV (2.775 A) was selected as this energy provides a good balance of increased signal from P and S atoms and reduced X-ray absorption by the crystal and solvent. Each data set consisted of 360° with an exposure time of 0.1 s per 0.1° image. Multiple data sets per crystal were taken at varying kappa and phi values to ensure completeness and increase multiplicity. Data integration was performed with XDS and XSCALE (Kabsch, 2010). The origin of the PDB model was corrected using POINTLESS, AIMLESS and MOLREP (Evans, 2005; Winn *et al*, 2011). Phased anomalous difference Fourier maps were produced using ANODE (Thorn & Sheldrick, 2011). A sigma cut-off of 4.0 was used to identify sites of anomalous contribution.

## Molecular graphics

Molecular analyses and structure alignments were performed with the UCSF Chimera package (Pettersen *et al*, 2004). Chimera is developed by the Resource for Biocomputing, Visualization, and Informatics at the University of California, San Francisco (supported by NIGMS P41-GM103311). Molecular graphics were produced in PyMOL Molecular Graphics System, version 2.3.3 Schrödinger, LLC.

## Data availability

The NMR assignments were deposited to the BMRB database (http://www.bmrb.wisc.edu/) and assigned the accession number BMRB ID: 50602. The Mad1$^{CTD}$-Bub1$^{CD1}$ coordinates have been deposited to the PDB with the identification codes, 7B1F (P2$_1$2$_1$2$_1$), 7B1H (P21) and 7B1J (P2$_1$2$_1$2).

**Expanded View** for this article is available online.

## Acknowledgements
E.S.F. was funded by a Gates Cambridge Scholarship. This work was funded by MRC grant (MC_UP_1201/6) and CRUK grant (C576/A14109) to D. Barford. We thank all current members of DB group and previous members, C. Alfieri and A. Boland, as well as L. Kiss for helpful discussions and experimental advice. We thank beamline scientists at I23 and I04, Diamond Light source, Didcot, UK, for help with crystallographic data collection. We also thank F. Gorrec for advice regarding 4-corner crystallization optimization and M. Skehel and S. Maslen for mass spectrometry analysis. All NMR data were acquired at the MRS facility of the MRC-LMB.

## Author contributions
ESF and DBa conceived the project. ESF cloned, isotopically labelled, purified all proteins and performed ITC experiments. ESF and DBe crystallized and collected Mad1 X-ray data. ESF with help from DBe and DBa solved the X-ray structures and refined the models. CWHY and SMVF analysed NMR data. SHM performed and analysed analytical ultracentrifugation and SEC-MALS experiments. AW and CMO completed SAD experiments. ESF, CWHY and DBa wrote the manuscript with inputs from all authors.

## Conflict of interest
The authors declare that they have no conflict of interest.

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
