## [Review Process File · EMBO Reports]

Molecular mechanism of Mad1 kinetochore targeting by phosphorylated Bub1

Elyse Fischer, Conny Yu, Dom Bellini, Stephen McLaughlin, Christian Orr, Armin Wagner, Stefan Freund, and David Barford

DOI: [10.15252/embr.202052242](https://doi.org/10.15252/embr.202052242)

Corresponding author(s): David Barford (dbarford@mrc-lmb.cam.ac.uk)

Review Timeline:

Submission Date:	9th Dec 20
Editorial Decision:	19th Jan 21
Revision Received:	24th Feb 21
Editorial Decision:	30th Mar 21
Revision Received:	2nd Apr 21
Accepted:	13th Apr 21

Editor: Deniz Senyilmaz Tiebe

Transaction Report:

Dear Dr. Barford,

Thank you for the submission of your research manuscript to our journal, which was now seen by three referees, whose reports are copied below.

I apologize for this unusual delay in getting back to you. It took longer than anticipated to receive the full set of referee reports due to the recent holiday season.

Referees express interest in the analysis. However, they also raise overlapping important concerns that need to be addressed to consider publication here.

I find the reports informed and constructive, and believe that addressing the concerns raised will significantly strengthen the manuscript. As the reports are below, and I think all points need to be addressed, I will not detail them here.

As for the 9th point of referee #3, while we advise that text and figures should be as concise as possible, there are no outright character count limitations for the Research Article format. However, Research Articles should have at least 6 main figures. To keep the manuscript as an Research Article, please convert one supplementary figure into a main figure.

Given these positive recommendations, we would like to invite you to revise your manuscript with the understanding that the referee concerns (as in their reports) must be fully addressed and their suggestions taken on board. Please address all referee concerns in a complete point-by-point response. Acceptance of the manuscript will depend on a positive outcome of a second round of review. It is EMBO reports policy to allow a single round of revision only and acceptance or rejection of the manuscript will therefore depend on the completeness of your responses included in the next, final version of the manuscript.

*** Temporary update to EMBO Press scooping protection policy:

We are aware that many laboratories cannot function at full efficiency during the current COVID-19/SARS-CoV-2 pandemic and have therefore extended our 'scooping protection policy' to cover the period required for a full revision to address the experimental issues highlighted in the editorial decision letter. Please contact the scientific editor handling your manuscript to discuss a revision plan should you need additional time, and also if you see a paper with related content published elsewhere.***

IMPORTANT NOTE: we perform an initial quality control of all revised manuscripts before re-review. Your manuscript will **FAIL** this control and the handling will be **DELAYED** if the following **APPLIES**:

1. A data availability section providing access to data deposited in public databases is missing (where applicable).
2. Your manuscript contains statistics and error bars based on $n=2$. Please use scatter plots in

these cases.

You can submit the revision either as a Scientific Report or as a Research Article. For Scientific Reports, the revised manuscript can contain up to 5 main figures and 5 Expanded View figures. If the revision leads to a manuscript with more than 5 main figures it will be published as a Research Article. In this case the Results and Discussion section can stay as it is now. If a Scientific Report is submitted, these sections have to be combined. This will help to shorten the manuscript text by eliminating some redundancy that is inevitable when discussing the same experiments twice. In either case, all materials and methods should be included in the main manuscript file

Supplementary/additional data: The Expanded View format, which will be displayed in the main HTML of the paper in a collapsible format, has replaced the Supplementary information. You can submit up to 5 images as Expanded View. Please follow the nomenclature Figure EV1, Figure EV2 etc. The figure legend for these should be included in the main manuscript document file in a section called Expanded View Figure Legends after the main Figure Legends section. Additional Supplementary material should be supplied as a single pdf labeled Appendix. The Appendix includes a table of content on the first page with page numbers, all figures and their legends. Please follow the nomenclature Appendix Figure Sx throughout the text and also label the figures according to this nomenclature. For more details please refer to our guide to authors.

Please note that for all articles published beginning 1 July 2020, the EMBO Reports reference style will change to the Harvard style for all article types. Details and examples are provided at <https://www.embopress.org/page/journal/14693178/authorguide#referencesformat>

2) individual production quality figure files as .eps, .tif, .jpg (one file per figure).

3) a .docx formatted letter INCLUDING the reviewers' reports and your detailed point-by-point responses to their comments. As part of the EMBO Press transparent editorial process, the point-by-point response is part of the Review Process File (RPF), which will be published alongside your paper. For more details on our Transparent Editorial Process, please visit our website: <https://www.embopress.org/page/journal/14693178/authorguide#transparentprocess>
You are able to opt out of this by letting the editorial office know (emboreports@embo.org). If you do opt out, the Review Process File link will point to the following statement: "No Review Process File is available with this article, as the authors have chosen not to make the review process public in this case."

4) a complete author checklist, which you can download from our author guidelines (<http://embor.embopress.org/authorguide>). Please insert information in the checklist that is also reflected in the manuscript. The completed author checklist will also be part of the RPF.

5) Please note that all corresponding authors are required to supply an ORCID ID for their name upon submission of a revised manuscript (<https://orcid.org/>). Please find instructions on how to link your ORCID ID to your account in our manuscript tracking system in our Author guidelines (<http://embor.embopress.org/authorguide>).

6) We replaced Supplementary Information with Expanded View (EV) Figures and Tables that are collapsible/expandable online. A maximum of 5 EV Figures can be typeset. EV Figures should be cited as 'Figure EV1, Figure EV2' etc... in the text and their respective legends should be included in the main text after the legends of regular figures.

- For the figures that you do NOT wish to display as Expanded View figures, they should be bundled together with their legends in a single PDF file called *Appendix*, which should start with a short Table of Content. Appendix figures should be referred to in the main text as: "Appendix Figure S1, Appendix Figure S2" etc. See detailed instructions regarding expanded view here: <http://embor.embopress.org/authorguide#expandedview>.

7) We would also encourage you to include the source data for figure panels that show essential data.

Numerical data should be provided as individual .xls or .csv files (including a tab describing the data). For blots or microscopy, uncropped images should be submitted (using a zip archive if multiple images need to be supplied for one panel). Additional information on source data and instruction on how to label the files are available <http://embor.embopress.org/authorguide#sourcedata>.

8) Our journal encourages inclusion of *data citations in the reference list* to directly cite datasets that were re-used and obtained from public databases. Data citations in the article text are distinct from normal bibliographical citations and should directly link to the database records from which the data can be accessed. In the main text, data citations are formatted as follows: "Data ref: Smith et al, 2001" or "Data ref: NCBI Sequence Read Archive PRJNA342805, 2017". In the Reference list, data citations must be labeled with "[DATASET]". A data reference must provide the database name, accession number/identifiers and a resolvable link to the landing page from which the data can be accessed at the end of the reference. Further instructions are available at <http://embor.embopress.org/authorguide#datacitation>.

9) Please make sure to include a Data Availability Section before submitting your revision - if it is not applicable, make a statement that no data were deposited in a public database. Primary datasets (and computer code, where appropriate) produced in this study need to be deposited in an appropriate public database (see <http://embor.embopress.org/authorguide#dataavailability>).

The accession numbers and database should be listed in a formal "Data Availability " section (placed after Materials & Method) that follows the model below. Please note that the Data Availability Section is restricted to new primary data that are part of this study.

Data availability

10) Regarding data quantification, please ensure to specify the name of the statistical test used to generate error bars and P values, the number (n) of independent experiments underlying each data point (not replicate measures of one sample), and the test used to calculate p-values in each figure legend. Discussion of statistical methodology can be reported in the materials and methods section, but figure legends should contain a basic description of n, P and the test applied. Please note that error bars and statistical comparisons may only be applied to data obtained from at least three independent biological replicates. Please also include scale bars in all microscopy images.

I look forward to seeing a revised version of your manuscript when it is ready. Please let me know if you have questions or comments regarding the revision.

Yours sincerely,

Deniz Senyilmaz Tiebe

Deniz Senyilmaz Tiebe, PhD
Editor
EMBO Reports

Referee #1:

In this work from Barford and co-workers the structure of the Mad1-Bub1 complex is determined and analyzed. Mad1 is a fully evolutionary conserved component of the spindle assembly checkpoint (SAC) a checkpoint that monitors kinetochore-microtubule interactions during mitosis. For a functional SAC it is crucial that Mad1 binds the checkpoint component Bub1. The RLK motif in Mad1 and a conserved domain (CD1) in Bub1 have been shown to be required for this interaction which only occurs when CD1 is phosphorylated on Thr461 by Mps1. However, the structural basis for the interaction is missing and represents a major gap in our understanding of the SAC. Here the authors solve the crystal structure of a Mad1 C-terminal fragment with a Bub1 CD1 peptide in its phosphorylated state. The structure reveals that Thr461-P interacts with the Arg residue of the RLK motif in Mad1 consistent with the existing biochemical and cellular data. If furthermore identifies a possible function of Thr461 phosphorylation in acting as a cap on the helical dipole which could be important for Mad1-Bub1 interaction. Finally, additional interactions between CD1 and Mad1 likely contributes to binding. In the structure the authors identify two CD1 peptides

binding to the Mad1 dimer while a panel of binding assays they conduct reveal that in solution the Mad1 dimer only binds one Bub1 CD1 peptide. The binding of two CD1 peptides to Mad1 is likely the result of high protein concentrations and the presence of crystallographic additives and not an artefact from crystal packing.

Overall, the work is of high quality and is well written. Given that the structure they elucidate is of fundamental importance for understanding checkpoint signaling across eukaryotes I recommend its publication in EMBO Reports and only have minor comments.

- The authors conduct a number of experiments to argue that the Mad1 dimer only binds one CD1 peptide and that this is likely the stoichiometry in vivo. This appears to be due to an asymmetry in the Mad1 dimer. However, it is important to remember that this is not done with full length proteins so one has to be cautious in conclusions and I think this caveat should be pointed out. In relation to this could the authors try and estimate concentrations of Mad1 and Bub1 at kinetochores which could help establish if concentrations could be high enough allowing two Bub1 molecules to bind Mad1. The authors should also comment on observations from yeast suggesting a different stoichiometry (PMID: 27170178)
- Have the authors analyzed if Mad1 can bind directly to Cdc20 and is this influenced by Bub1 CD1 as proposed in Ji et al 2017? This would have important implications for understanding SAC signaling also in light of the two recent Science papers on this topic. The authors are focused on the model that Bub1 positions Cdc20 close to Mad1 but this model is not consistently supported by the literature (see for example: Di Fiore Dev Cell 2016 where a Cdc20 mutant unable to bind ABBA motifs still loads Mad2 and Zhang et al 2019, PMID: 30782962 where Bub1 without an ABBA motif still supports a robust checkpoint).
- In figure 3B a small schematic of the Mad1-Bub1 CD1 complex and the mutations/modifications they test in ITC would be helpful. Looking at the table there is a lot of information that is not easy to follow unless you look at the other figures.
- In figure 5A could the authors indicate the peaks they want the readers to focus on that are also discussed in the text.

Referee #2:

The spindle assembly checkpoint (SAC) ensures the accuracy of chromosome segregation to prevent aneuploidy. Through a sequential phosphorylation-dependent signaling cascade, the SAC master kinase, MPS1, promotes the direct interaction between the BUB1 central conserved domain 1 (CD1) and the MAD1 C-terminal domain (CTD). This BUB1-MAD1 interaction helps to build a scaffold for the assembly of the mitotic checkpoint complex (MCC), an inhibitor of the anaphase promoting complex/cyclosome (APC/C), to delay anaphase onset until proper kinetochore-microtubule attachments. In this manuscript, Fischer et al. presents the high-resolution crystal structure of human MAD1 CTD bound to a phospho-BUB1 CD1 peptide. The structure reveals that pThr461 (but not pSer459) in BUB1 CD1 makes direct contact with the conserved RLK motif in MAD1 CTD. Through mutagenesis analysis, ITC assays, and NMR spectroscopy, they show that only one BUB1 CD1 molecule binds to one MAD1 CTD homodimer. Finally, they provide evidence to suggest that upon BUB1 CD1 binding, MAD1 CTD undergoes conformational changes in solution.

Overall, this manuscript provides the first detailed structural characterization of a key regulated interaction between BUB1 and MAD1, which is crucial for checkpoint activation. The biochemical results are convincing. Publication of this excellent study in EMBO Reports is highly recommended, provided that the following minor points are addressed.

Minor points:

- 1) On page 2, the first line of the third paragraph, Msp1 should be Mps1.
- 2) Please provide a figure to show the omit map (2Fo–Fc map) for the BUB1 CD1 peptide.

Referee #3:

Kinetochores signaling during the spindle assembly checkpoint requires the assembly of a Mad1-Mad2 complex at the kinetochore. Their recruitment is mediated by binding of an RLK motif within the C-terminal domain of Mad1 to a phosphopeptide motif within Bub1. The authors analyze the interaction between the Mad1-CTD and a Bub1 peptide using X-ray crystallography, with follow-up analysis using a variety of biophysical methods.

Quite a bit was already known about the details of this interaction, so much of the basic structure, while adding atomic detail, is largely confirmatory. What is new is strong evidence that only one of two phosphorylations of this region of Bub1 (P-Thr-461) participates in binding Mad1-CTD, contrary to previous reports, and the finding that the two molecules forming the Mad1-CTD dimer adopt different structures, and that only one Bub1 peptide binds per Mad1-CTD dimer in solution, despite two molecules of the Bub1 peptide binding in the crystal structure. Despite being based on just a portion of Mad1 and a peptide from Bub1, these interesting new findings are strongly supported and likely to reflect the mechanism involving full-length proteins.

Comments:

- 1) p. 2, middle. "Msp1" should be "Mps1."
- 2) p. 5, near bottom. The authors state that the singly phosphorylated Bub1 peptide (on Thr-461) is less soluble than the double phosphorylated peptide (also phosphorylated on Ser-459), which may explain previous reports that P-Ser-459 contributes to binding Mad1-CTD. It would be helpful if the authors could quantify this reduced solubility.
- 3) p. 10, near top. The authors suggest that high peptide concentration plus the presence of DMSO and isopropanol might explain the presence of two Bub1 peptides bound to each Mad1-CTD in the crystal structure. Could this prediction be tested in solution (at least involving the solvents)? Could crystal packing also have contributed? Relatedly, have the authors ever tried titrating down the concentration of the Bub1 peptide to try and obtain crystals with one molecule of Bub1 per Mad1-CTD dimer? That structure might be very informative.
- 4) p. 10, near bottom. It was unclear whether the asymmetry in the Mad1-CTD dimer was pointed out in previous studies, or if its subtlety was only noticed in the current study.
- 5) p. 10, near bottom. It would be helpful to point out (or at least foreshadow) the Mad1-CTD subunit asymmetry earlier in the presentation. Until this point, I had assumed that the Mad1-CTD contained symmetric subunits and that the binding of one Bub1 peptide changed the structure of the other MAD1-CTD so that it could no longer bind peptide. An earlier understanding that the asymmetry preceded Bub1 binding would aid the reader.

6) Materials and Methods. Please cite publications, rather than manufacturers, for specific methods. This issue was particularly noticeable in the Cloning... section.

7) Figure 3C legend. Please comment on the bottom (1:2) spectrum.

8) Supplemental Figure 7A. It's not clear if the Mad1-CTD profile is shown in the figure (obscured by F629A?). In the Figure, as L618A mislabeled as 26 kDa rather than 29.6 kDa (as stated in the legend)? Should this figure be included in the primary section? It seems fairly important.

9) I'm unclear if the manuscript satisfies the journal's length requirements. I raised this issue prior to review and was told "The 25,000 character limit is for our shorter format Report style papers. This paper is a Research Article and falls within the scope of our journal format." While filling out the review, the website text said that "The present version of the paper has a length of 24990 characters - 25,000 characters are allowed." However, by my count, just the Introduction plus the Results and Discussion section is nearly 40,000 characters (including spaces).

Reviewers' comments:

Referees' comments in blue, changes to manuscript text in red.

Referee #1:

We thank the reviewer for their positive and constructive comments.

1-1) The authors conduct a number of experiments to argue that the Mad1 dimer only binds one CD1 peptide and that this is likely the stoichiometry in vivo. This appears to be due to an asymmetry in the Mad1 dimer. However, it is important to remember that this is not done with full length proteins so one has to be cautious in conclusions and I think this caveat should be pointed out.

Response: We agree with the reviewer. Using the Bub1^{CD1} and Mad1^{CTD} truncations is a limitation to our study. We have modified the text to acknowledge this: pages 9 and 10.

In our original manuscript, we had tested the binding of the Bub1^{CD1} peptide to Mad1⁴⁸⁵⁻⁷¹⁸ in complex with Mad2 that forms the tetrameric Mad1:C-Mad2 complex (**Appendix Figure S4D**). We have now also tested Bub1^{CD1} binding to a longer version of Mad1, Mad1⁴²⁰⁻⁷¹⁸ in complex with Mad2 (**Appendix Figure S4E**). In both complexes the stoichiometry is maintained, although the K_D slightly increased. Because this stoichiometry is conserved in the Mad1-Mad2 tetrameric complex, it likely represents the physiological stoichiometry in cells. However as mentioned above we cannot exclude the possibility of two Bub1 molecules interacting with the Mad1-Mad2 complex in cells. We now discuss in the text: page 9.

1-2) In relation to this could the authors try and estimate concentrations of Mad1 and Bub1 at kinetochores which could help establish if concentrations could be high enough allowing two Bub1 molecules to bind Mad1.

Response: This is an interesting point. Several prior studies have determined the concentrations of Mad1 and Bub1 at kinetochores. We have therefore not performed additional experiments.

Experiments which quantified kinetochore levels of Mad1 and Bub1 were summarised in the supplementary material of the Faesen *et al.*, 2017 paper, "Supplementary Section B: Cellular Concentrations of Checkpoint Proteins" (Faesen *et al.*, 2017). There the authors cite several papers, indicating Bub1 at 100 nM and Mad1 at 20 nM or 25% of Mad2, and Mad2 at 120-400 nM (Howell *et al.*, 2004, 2000; Tang *et al.*, 2001; Sudakin *et al.*, 2001; Fang, 2002; Luo *et al.*, 2004; Shah *et al.*, 2004). We have now mentioned this in the text on page 10.

1-3) The authors should also comment on observations from yeast suggesting a different stoichiometry (PMID: 27170178)

Response: Thank you for pointing out this paper. We have now mentioned this observation in the text: bottom of page 10.

2-1) Have the authors analyzed if Mad1 can bind directly to Cdc20 and is this influenced by Bub1 CD1 as proposed in Ji et al 2017? This would have important implications for understanding SAC signaling also in light of the two recent Science papers on this topic.

Response: We have not carefully analysed the Mad1-Cdc20 interaction. We have replicated the report from Ji et al., 2017, that Cdc20 has very weak binding (only detectable by western blotting) to Mad1^{CTD} (data not shown). Additionally, we found that phosphorylated Bub1-Bub3 does not detectably enhance the Mad1-Cdc20 interaction.

2-2) The authors are focused on the model that Bub1 positions Cdc20 close to Mad1 but this model is not consistently supported by the literature (see for example: Di Fiore Dev Cell 2016 where a Cdc20 mutant unable to bind ABBA motifs still loads Mad2 and Zhang et al 2019, PMID: 30782962 where Bub1 without an ABBA motif still supports a robust checkpoint).

Response: Thank you for pointing this out. We agree that we predominately focused on the model that the Bub1 ABBA/KEN motifs position Cdc20 close to Mad1, and that we should have stated more clearly that this model is still debated.

Two changes to the text were made:

- 1) In the Introduction (page 2) where we had previously already discussed how this model is debated, we added the (Zhang *et al*, 2019) reference.
- 2) In the discussion of the Bub1-Mad1-Cdc20 complex at kinetochores at the bottom of page 4 we mention once again that this model is debated.

3) In figure 3B a small schematic of the Mad1-Bub1 CD1 complex and the mutations/modifications they test in ITC would be helpful. Looking at the table there is a lot of information that is not easy to follow unless you look at the other figures.

Response: We agree, thank you. We have added a schematic (**Fig 3C**) and updated the figure legend which we hope will help interpret the ITC data more easily.

4) In figure 5A could the authors indicate the peaks they want the readers to focus on that are also discussed in the text.

Response: In **Figure 5**, we have now highlighted in red peaks of interest. We added two additional figures to the appendix (**Appendix Fig S7** and **Appendix Fig S8**) which highlight the chemical perturbations of these peaks during Bub1^{CD1} titration. Additionally, all peaks

which are affected are already mapped out in **Fig 5B**, and this can be tracked across Bub1^{CD1} titration in **Fig EV5**.

Referee #2:

We thank the reviewer for their positive and constructive comments.

Minor points:

1) On page 2, the first line of the third paragraph, Msp1 should be Mps1.

Response: Thank you, we have corrected this error.

2) Please provide a figure to show the omit map (2Fo-Fc map) for the BUB1 CD1 peptide.

Response: We have created a 2Fo-Fc omit map and included it in **Fig EV2D** and updated the figure legend accordingly.

Referee #3:

We thank the reviewer for their positive and constructive comments.

Comments:

1) p. 2, middle. "Msp1" should be "Mps1."

Response: Thank you, we have corrected this error.

2) p. 5, near bottom. The authors state that the singly phosphorylated Bub1 peptide (on Thr-461) is less soluble than the double phosphorylated peptide (also phosphorylated on Ser-459), which may explain previous reports that P-Ser-459 contributes to binding Mad1-CTD. It would be helpful if the authors could quantify this reduced solubility.

Response: We realise that our comment on the reduced solubility of the singly phosphorylated pT461 peptide is likely confusing, so we have removed this comment.

In terms of quantifying the reduced solubility, we observed an approximately 2-fold reduction of the solubility of the singly phosphorylated peptide. The pS459-pT461 peptide was soluble in 20 mM Hepes, pH 7.5, 100 mM NaCl, 1 mM TCEP, without DMSO, to 1.5 mM, whereas the singly phosphorylated pT461 peptide was only soluble to about 0.7 mM. Therefore, as we usually performed our ITC experiments with 100 μ M Mad1^{CTD} and titrated 1 mM Bub1^{CD1} pS459-pT461, we lowered the concentration of the singly phosphorylated pT461 peptide to 0.5 mM and reduced Mad1^{CTD} to 50 μ M. Lowering the concentration of the doubly phosphorylated peptide experiment resulted in the same K_D , thus in our hands there

seems to be no difference in binding between the singly and doubly phosphorylated peptides under these conditions *in vitro*.

3-1) p. 10, near top. The authors suggest that high peptide concentration plus the presence of DMSO and isopropanol might explain the presence of two Bub1 peptides bound to each Mad1-CTD in the crystal structure. Could this prediction be tested in solution (at least involving the solvents)?

Response: This is an important point which we previously explored. However, using DMSO/isopropanol in our ITC experiments was not possible, because the solvents, particularly DMSO, caused large enthalpy changes. The ^1H NMR signal from DMSO obscures the protein peaks. However, NMR is able to detect weak interactions in the mM range, and our NMR titrations as well as the ^{31}P NMR experiments, showed no indications for a second weaker binding site present on the Mad1^{CTD} dimer, even with almost millimolar concentrations of Mad1^{CTD} and Bub1^{CD1}. This indicates that the binding of the second peptide observed in the crystal is mostly likely caused by the solvent and/or crystallisation process. This is now discussed on page 10.

3-2) Could crystal packing also have contributed?

Response: We think not. In all three Mad1^{CTD}-Bub1^{CD1} crystal structures (P2₁, P2₁2₁2₁, P2₁2₁2₁), in which the crystal packing is substantially varied, there are no clear crystallographic contacts with either of the peptides. Discussed on page 12.

3-3) Relatedly, have the authors ever tried titrating down the concentration of the Bub1 peptide to try and obtain crystals with one molecule of Bub1 per Mad1-CTD dimer? That structure might be very informative.

Response: This is an excellent suggestion and is something we had attempted. In our hands, lowering the peptide and/or DMSO concentration resulted in either an increase in the differential occupancy of the peptides (the lower occupancy peptide had even lower occupancy) or crystallisation of apo Mad1^{CTD}. We suspect one reason for this is that in the apo Mad1^{CTD} structure, the P6 space group has very tight packing of the Mad1^{CTD} dimers that blocks the binding site of the peptide. Attempts to soak the Bub1^{CD1} peptide at high concentration into the apo Mad1^{CTD} crystals, even in the presence of DMSO were not successful. We assume that at lower concentrations of the peptide, the tight P6 packing of the apo Mad1^{CTD} is favoured. Discussed on page 10.

4) p. 10, near bottom. It was unclear whether the asymmetry in the Mad1-CTD dimer was pointed out in previous studies, or if its subtlety was only noticed in the current study.

Response: Thank you for pointing this out. To our knowledge, we are the first to comment on the asymmetry of Mad1^{CTD}. We have now added a statement to point this out in the text. Additionally, the crystal structure of tetrameric human Mad1:C-Mad2 complex from the

Musacchio lab (Sironi *et al*, 2002), was pointed out to be significantly asymmetric, and so we have now also added a brief discussion of this in the text as well: Page 12.

5) p. 10, near bottom. It would be helpful to point out (or at least foreshadow) the Mad1-CTD subunit asymmetry earlier in the presentation. Until this point, I had assumed that the Mad1-CTD contained symmetric subunits and that the binding of one Bub1 peptide changed the structure of the other MAD1-CTD so that it could no longer bind peptide. An earlier understanding that the asymmetry preceded Bub1 binding would aid the reader.

We already mentioned the asymmetry in both the abstract and introduction. To strengthen this we add: ‘also apparent in the previously crystallized apo Mad1^{CTD} homodimer (Kim *et al*, 2012)’: Page 3.

6) Materials and Methods. Please cite publications, rather than manufacturers, for specific methods. This issue was particularly noticeable in the Cloning... section.

Response: Thank you for bringing this to our attention. We now cite the specific methods used in our cloning as well as a few other methods described in the text: pages 15 and 17.

7) Figure 3C legend. Please comment on the bottom (1:2) spectrum.

Response: We have added this comment to the legend of what is now **Figure 3D**. “In the 1:2 molar ratio of Mad1^{CTD} dimer to Bub1^{CD1}, in addition to signal for the bound pSer459, there is reappearance of the original free position of pS459 and unbound pThr461 supporting the presence of unbound peptide.”

8-1) Supplemental Figure 7A. It's not clear if the Mad1-CTD profile is shown in the figure (obscured by F629A?).

Response: The black and pink line of Mad1^{CTD} WT and F629A correspondingly, do indeed overlap, so we have changed the thickness of the lines so that both are now more visible to improve clarity.

8-2) In the Figure, as L618A mislabeled as 26 kDa rather than 29.6 kDa (as stated in the legend)?

Response: Thank you, we have corrected this mislabelling in the figure. It is indeed 29.6 kDa as stated in the figure legend.

8-3) Should this figure be included in the primary section? It seems fairly important.

Response: The main reason we performed and included the SEC-MALS experiments was to check that these Mad1^{CTD} mutants, which are positioned at the hydrophobic dimerization interface of Mad1^{CTD}, were still dimeric. Although they were dimeric, we suspect that the

reason why the yield was significantly reduced, and the mutants were more prone to aggregation, was because they are partially misfolded. However, because we do not have conclusive evidence that F629A/L618A are misfolded dimers, we do not think this figure warrants being in the primary section. We have therefore decided not to include this figure in the primary figures but instead as an Extended View figure which makes it still easily accessible to readers.

9) I'm unclear if the manuscript satisfies the journal's length requirements. I raised this issue prior to review and was told "The 25,000 character limit is for our shorter format Report style papers. This paper is a Research Article and falls within the scope of our journal format." While filling out the review, the website text said that "The present version of the paper has a length of 24990 characters - 25,000 characters are allowed." However, by my count, just the Introduction plus the Results and Discussion section is nearly 40,000 characters (including spaces).

Response: We apologise for this confusion. Our manuscript will be published as an EMBO Reports Research Article, which does not have a word limit. To fully satisfy publication as a Research Article, as suggested by the editor, we have converted the original Supp Figure 1 into a main figure, Figure 6.

Dear Dr. Barford

Thank you for submitting your revised manuscript. It has now been seen by two of the original referees. As you can see, the referees find that the study is significantly improved during revision. Before I can accept the manuscript, I need you to address the additional points below:

- Please change the title of the section Declaration of Interests to Conflict of Interest.
- In the Author Contributions section, please abbreviate D Barford as DBa and D Bellini as DBe.
- We noted the following regarding the figure callouts:
 - Fig 1 panels are not called out.
 - Fig EV2C-G callouts are missing.
 - Fig EV4A&B callouts are missing.
 - Fig EV5 panels callouts are missing.
 - Appendix Fig S3 panel callouts are missing.
 - Appendix Fig S6A callout is missing.
- The callout format of Appendix is Appendix Table SX or Appendix Figure SX. We note that current the S letter is missing from the Appendix callouts.
- Please upload the Appendix file as pdf.
- Please make the NMR assignments deposited to the BMRB database publicly available.
- Papers published in EMBO Reports include a 'synopsis' and 'bullet points' to further enhance discoverability. Both are displayed on the html version of the paper and are freely accessible to all readers. The synopsis includes a short standfirst summarizing the study in 1 or 2 sentences that summarize the paper and are provided by the authors and streamlined by the handling editor. I would therefore ask you to include your synopsis blurb and 3-5 bullet points listing the key experimental findings.
- In addition, please provide an image for the synopsis. This image should provide a rapid overview of the question addressed in the study but still needs to be kept fairly modest since the image size cannot exceed 550x400 pixels.

Thank you again for giving us to consider your manuscript for EMBO Reports, I look forward to your minor revision.

Kind regards,

Deniz Senyilmaz Tiebe

--

Deniz Senyilmaz Tiebe, PhD
Editor
EMBO Reports

Referee #1:

The authors have addressed all my points and improved the manuscript based on all comments from the reviewers. I support publication in EMBO Reports.

Jakob Nilsson

Referee #2:

The authors have addressed my minor concerns raised during the previous round of reviews. Publication of this excellent manuscript is highly recommended.

The authors have addressed all minor editorial requests.

Dear David,

Thank you for submitting your revised manuscript. I have now looked at everything and all is fine. Therefore, I am very pleased to accept your manuscript for publication in EMBO Reports.

Congratulations on a nice work!

Kind regards,

Deniz

--

Deniz Senyilmaz Tiebe, PhD
Editor
EMBO Reports

--

At the end of this email I include important information about how to proceed. Please ensure that you take the time to read the information and complete and return the necessary forms to allow us to publish your manuscript as quickly as possible.

As part of the EMBO publication's Transparent Editorial Process, EMBO reports publishes online a Review Process File to accompany accepted manuscripts. As you are aware, this File will be published in conjunction with your paper and will include the referee reports, your point-by-point response and all pertinent correspondence relating to the manuscript.

If you do NOT want this File to be published, please inform the editorial office within 2 days, if you have not done so already, otherwise the File will be published by default [contact: emboreports@embo.org]. If you do opt out, the Review Process File link will point to the following statement: "No Review Process File is available with this article, as the authors have chosen not to make the review process public in this case."

Should you be planning a Press Release on your article, please get in contact with emboreports@wiley.com as early as possible, in order to coordinate publication and release dates.

Thank you again for your contribution to EMBO reports and congratulations on a successful publication. Please consider us again in the future for your most exciting work.

Yours sincerely,

Deniz Senyilmaz Tiebe, PhD
Editor
EMBO Reports

THINGS TO DO NOW:

You will receive proofs by e-mail approximately 2-3 weeks after all relevant files have been sent to our Production Office; you should return your corrections within 2 days of receiving the proofs.

Please inform us if there is likely to be any difficulty in reaching you at the above address at that time. Failure to meet our deadlines may result in a delay of publication, or publication without your corrections.

All further communications concerning your paper should quote reference number EMBOR-2020-52242V3 and be addressed to emboreports@wiley.com.

Should you be planning a Press Release on your article, please get in contact with emboreports@wiley.com as early as possible, in order to coordinate publication and release dates.

Corresponding Author Name: Dr. David Barford

Manuscript Number: EMBOR-2020-52242V1